# A multiplex CRISPR interference tool for virulence gene interrogation in *Legionella pneumophila*

Nicole A. Ellis[1], Byoungkwan Kim[1], Jessica Tung[1] & Matthias P. Machner [1✉]

Catalytically inactive dCas9 imposes transcriptional gene repression by sterically precluding RNA polymerase activity at a given gene to which it was directed by CRISPR (cr)RNAs. This gene silencing technology, known as CRISPR interference (CRISPRi), has been employed in various bacterial species to interrogate genes, mostly individually or in pairs. Here, we developed a multiplex CRISPRi platform in the pathogen *Legionella pneumophila* capable of silencing up to ten genes simultaneously. Constraints on precursor-crRNA expression were overcome by combining a strong promoter with a *boxA* element upstream of a CRISPR array. Using crRNAs directed against virulence protein-encoding genes, we demonstrated that CRISPRi is fully functional not only during growth in axenic media, but also during macrophage infection, and that gene depletion by CRISPRi recapitulated the growth defect of deletion strains. By altering the position of crRNA-encoding spacers within the CRISPR array, our platform achieved the gradual depletion of targets that was mirrored by the severity in phenotypes. Multiplex CRISPRi thus holds great promise for probing large sets of genes in bulk in order to decipher virulence strategies of *L. pneumophila* and other bacterial pathogens.

[1] Division of Molecular and Cellular Biology, Eunice Kennedy Shriver National Institute of Child Health and Human Development, National Institutes of Health, Bethesda, MD 20892, USA. ✉email: machnerm@nih.gov

Clustered regularly interspaced short palindromic repeats (CRISPR)-Cas gene editing technologies have recently arisen as a mechanism for both fast and targeted gene manipulation in a variety of systems[1–3]. CRISPR-Cas-based genetic tools are derived from components of naturally occurring CRISPR systems found in 87% of archaea and 45% of bacteria genomes surveyed (Crisprfinder; http://crispr.i2bc.paris-saclay.fr/Server/). In these organisms, CRISPR regions serve as an adaptive immune system where fragments of foreign DNA elements such as viruses, transposons, or plasmids are incorporated into the chromosome as unique spacers (S) separated by identical repeats (R) to serve as a immunological memory of past infections[1,4,5]. RNA transcribed from these repeat/spacer regions, known as the precursor-CRISPR (cr)RNA, is processed into short fragments by RNase III to make individual crRNAs[6]. Upon re-infection, crRNAs, together with a trans-activating crRNA (tracrRNA), specifically direct, through base-pairing with the target, a protein (or contingent of proteins) with oligonucleotide cleavage capabilities to the foreign DNA or RNA elements for destruction[7–9]. Importantly, the spacers themselves are protected from self-targeting by the crRNAs that they encode as they are missing the protospacer-adjacent motif (PAM)[10], a short DNA element found directly downstream of the complementary target sequence on the non-target strand that is required by the surveillance complex to cut.

In the simplest CRISPR-Cas system, Type II, only a single protein known as Cas9 is required for crRNA-guided DNA cleavage, lending it to be the most developed as a genetic tool[11–13]. In bacteria, this system has been adapted for gene silencing by making two simple changes: First, the gene encoding Cas9 was replaced with a catalytically inactive variant of Cas9, called deactivated Cas9 (dCas9), in which the two nuclease domains, RuvC-like and HNH, have been mutated, thus preventing DNA cleavage[14,15]; and second, by designing arrays in which the spacer sequence(s), that are typically directed against invading DNA elements, have been replaced with sequences complementary to the bacterium's own genes. These self-targeting crRNAs form a surveillance complex together with the tracrRNA and with dCas9 which, in the absence of cleavage, imposes transcriptional gene repression by sterically precluding RNA polymerase activity at the gene to which the complex was directed, a technology referred to as CRISPR interference (CRISPRi)[14,16]. Gene silencing is advantageous to the study of essential genes that are otherwise intolerable to deletion, as well as for interrogating genes of interest without laborious null strain construction[17]. The recent development of mobile-CRISPRi has allowed for gene silencing to be performed in a number of gamma proteobacteria and Bacillales Firmicutes[18], yet this and earlier studies, which implemented CRISPRi[19], nearly always silenced only one gene or pairs of genes, with few exceptions[20–23]. To our knowledge, no group has fully exploited or probed the natural multiplex capability of CRISPR repeat/spacer arrays as a gene silencing tool in bacterial systems.

Here, we established an adaptable multiplex CRISPRi platform capable of silencing up to ten genes simultaneously (10-plex). We demonstrate that this platform can be applied to study bacterial virulence genes not only in axenic media but, importantly, also under disease-relevant conditions such as the infection of macrophages by the intracellular pathogen Legionella pneumophila, thus adding CRISPRi to our toolbox for efficiently studying genetically less tractable human pathogens.

## Results

### Equipping L. pneumophila with a CRISPRi system.
To study the potential of multiplex CRISPRi, we selected the model organism L. pneumophilia, the causative agent of Legionnaires'

pneumonia or a milder form of the disease called Pontiac fever[24,25]. The common laboratory strain L. pneumophila Philadelphia-1 (Lp02) does not bear an intrinsic CRISPR-Cas system (though some environmental L. pneumophila isolates do[26]). Thus, we exogenously introduced the genes encoding the three main components of a CRISPRi platform: a Cas protein, crRNAs, and the tracrRNA (Fig. 1a). The Streptococcus pyogenes dCas9-encoding sequence was inserted into the chromosomal thyA locus (lpg2868) as this L. pneumophila gene was already disrupted by a mutation in this strain background[27], creating Lp02(dcas9). Furthermore, dcas9 was placed under the control of the tetracycline-inducible tet promoter ($P_{tet}$) rather than its native S. pyogenes promoter, allowing the degree of gene repression to reflect the level of anhydrous tetracycline (aTC) added to the system to induce dcas9, as others have done before[14]. The crRNA-encoding repeat/spacer arrays were provided on plasmids, as was the target-independent (invariable) tracrRNA-encoding sequence. Construction of these single CRISPRi constructs is described in the Methods section and depicted in Supplementary Fig. S1a. Although single guide (sg)RNAs, chimeras of the tracrRNA and crRNA, are convenient for silencing individual genes as to circumvent the need for processing precursor-crRNA into a mature crRNAs[6], the tracrRNA-encoding sequences would become repetitive within longer CRISPR arrays. Instead, the task of processing crRNA precursors was seemingly executed efficiently by the intrinsic L. pneumophila RNAse III (lpg1869) in our system (see below).

Targeting individual genes with single crRNA arrays served as proof-of-concept for introducing a foreign CRISPR-Cas9 platform into our model organism L. pneumophila. These initial experiments were performed on L. pneumophila strains grown in axenic culture while individually targeting three virulence factor, or "effector", -encoding genes with crRNAs towards lidA (lpg0940; crRNA$^{lidA}$), sidM (lpg2464; crRNA$^{sidM}$), and vipD (lpg2831; crRNA$^{vipD}$), since antibodies directed against their encoded gene products were available. First, we confirmed that a gradual enhancement in the concentration of the inducer aTC led to increased levels of dCas9 and, consequently, decreased levels of LidA, the product of the target lidA, indeed creating a CRISPRi system capable of tunable gene silencing (Fig. 1b; note, all uncropped immunoblots are shown in Supplementary Fig. S8). Next, we assessed how the location of the crRNA annealing site on the target gene (strand, proximity to start codon) affects gene silencing efficiency. In agreement with other studies[12,14,28], we found that targeting a 30 base pair sequence near the start of the gene on the non-template strand provided the most efficient knockdown (Fig. 1c, arrowheads C and D). All target sequences were upstream of a PAM sequence ("NGG" on the template strand[29]).

Lastly, to confirm that crRNAs designed in this manner did not indiscriminately affect gene expression, we monitored protein (Fig. 1d) and mRNA (Fig. 1e) levels of unrelated genes. Using immunoblot analyses and quantitative polymerase chain reaction (qPCR), we found that, while the intended target genes' protein and mRNA levels were dramatically reduced by CRISPRi, none of the other surveyed genes chosen at random were repressed by crRNA expression. Together, these results confirmed that the aTC-inducible dcas9 and the plasmid-expressed crRNA and tracrRNA formed a functional CRISPRi surveillance complex in L. pneumophila that was tunable, efficient, and target-specific.

### A multiplex CRISPRi platform allows for the simultaneous repression of several genes.
Most processes in bacteria, including pathogenesis, rely on the combined activity of multiple gene products. With over 300 putative effectors, L. pneumophila

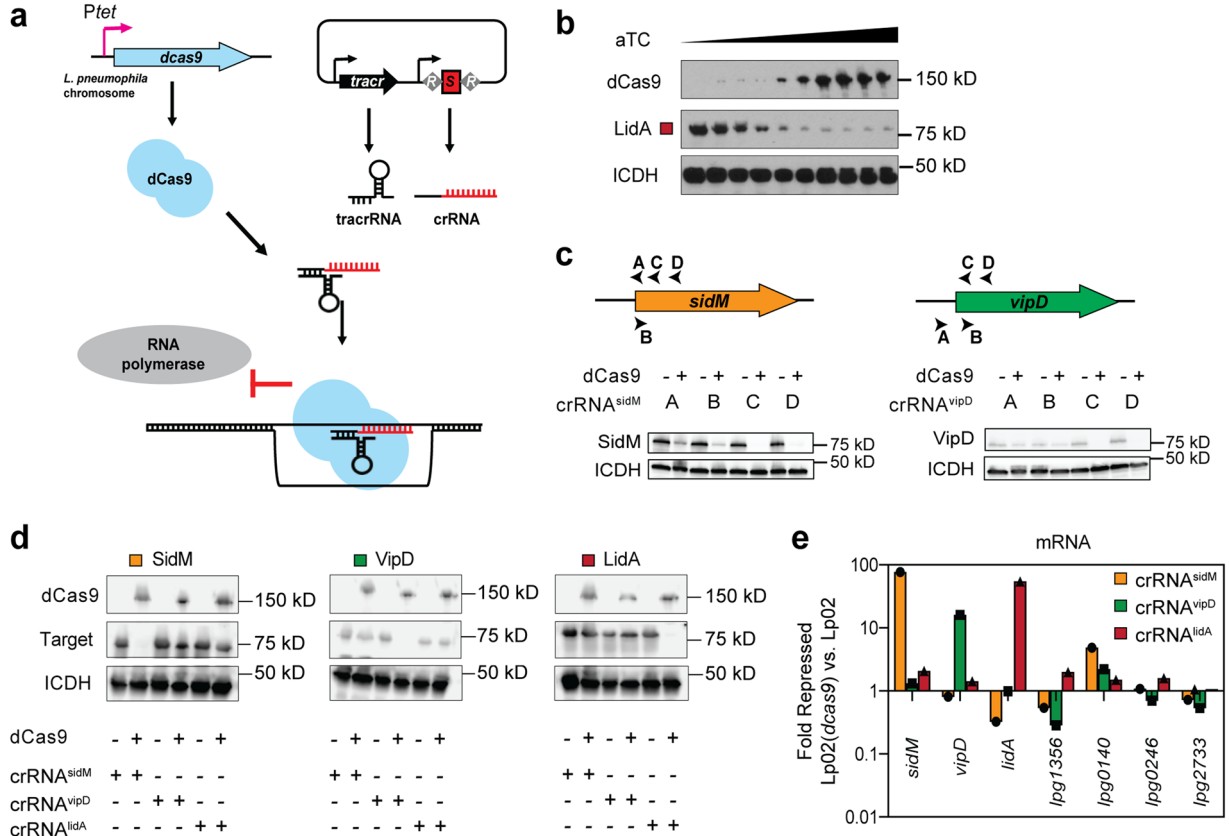

**Fig. 1 CRISPRi is adaptable to L. pneumophila. a** Schematic of the *L. pneumophila* single-plex CRISPRi platform. Chromosome-encoded dCas9 and plasmid-expressed tracrRNA and crRNA assemble into a CRISPRi complex. The crRNA directs the complex to the target gene through base pairing. Gene expression is repressed through sterically precluding RNA polymerase. R indicates repeats and S indicates spacers of the crRNA-encoding repeat/spacer array. **b** Immunoblot analysis showing the protein level of dCas9 in response to aTC-dependent induction of $P_{tet}$-*dcas9*, leading to decreased protein levels of the crRNA[lidA] target. Isocitrate dehydrogenase (ICDH) serves as a loading control. aTC concentrations are 0, 0.15, 0.3, 0.63, 1.23, 2.5, 5, 10, 20, and 80 ng/mL. **c** Immunoblots reveal the efficiency of target repression at different crRNA target sites (arrow heads A-D) for two genes, *sidM* and *vipD* (+40 ng/mL aTC). ICDH serves as a loading control. **d**, **e** Strains expressing crRNA[sidM] (site C), crRNA[vipD] (site C), and crRNA[lidA] were surveyed for CRISPRi specificity at both the protein (**d**) and mRNA (**e**) level (+40 ng/mL aTC). ICDH serves as an immunoblot loading control. qPCR was used to measure mRNA levels of *sidM, vipD, lidA* and four randomly chosen control genes in Lp02(*dcas9*) and Lp02 strains. Fold repression was determined using $\Delta\Delta C_T$. A value of 1 indicates that the gene level was the same in the Lp02(*dcas9*) and Lp02 strains.

encodes one of the largest known virulence arsenals of any bacterial pathogen[30,31]. We thus explored if our CRISPRi platform could be adapted to simultaneously repress more than one gene at a time. In this new multiplex CRISPRi format, expression of *S. pyogenes dcas9* from the *L. pneumophila* chromosome and the expression of the *S. pyogenes* tracrRNA from the plasmid backbone remained unchanged. Alterations were made to the method of crRNA expression: First, several unique crRNA-encoding spacer sequences, labeled *S1* (for the most proximal spacer) to *S10* (for the most distal), were sequentially placed between identical repeat sequences intrinsic to the *S. pyogenes* CRISPSR-Cas9 system (Fig. 2a). In nature, the number of spacers within a CRISPR array can vary, ranging from a few to several dozen or even hundreds[32]. Second, expression of the repeats and spacers was placed under the regulation of $P_{tet}$, the same aTC-inducible promoter used to express *dcas9* from the *L. pneumophila* chromosome. We had found in experiments on Lp02(*dcas9*) grown in axenic culture that using the native *S. pyogenes* promoter ($P_{native}$) to express the multiplex CRISPR (MC) repeat/spacer arrays proved insufficient in repressing more than three to four targets, possibly due to fewer incidences of transcription initiation at a weaker promoter and therefore, fewer opportunities for RNA polymerase to complete transcription of the entire array before

termination (Fig. 2b). Incorporation of $P_{tet}$ fostered expression of the array up to at least spacer *S9*, as confirmed by immunoblot analysis of the protein levels of LidA whose encoding gene was targeted by crRNAs expressed from the most distal spacer positions (Fig. 2b).

To examine the ability of our multiplex CRISPRi platform to target a variety of genes simultaneously, we first made a synthetic MC array, called MC-I, composed of ten spacers (10-plex) under the regulation of $P_{tet}$. The four spacers in position *S1*, *S3*, *S6*, and *S10* encoded crRNAs to target the effector-encoding genes *lidA*, *sidM*, *vipD*, and *ravN* (*lpg1111*), while the two spacers at position *S5* and *S8* target *dotD* (*lpg2674*) and *dotO* (*lpg0456*), two components of the type IV secretion system (T4SS) of *L. pneumophila*. Construction of MC constructs is described in the Methods section and depicted in Supplementary Fig. S1b. After growth of Lp02(*dcas9*) containing MC-I in axenic culture, we found that, indeed, the protein levels of all six targets were notably decreased based on immunoblot analysis as compared to a control strain bearing an empty vector (with tracrRNA but no repeat/spacer array) (Fig. 2c and Supplementary Fig. S2). This initial result showed that it is possible to simultaneously silence several unique genes in *L. pneumophila* using an array of repeats and spacers under the control of a strong promoter.

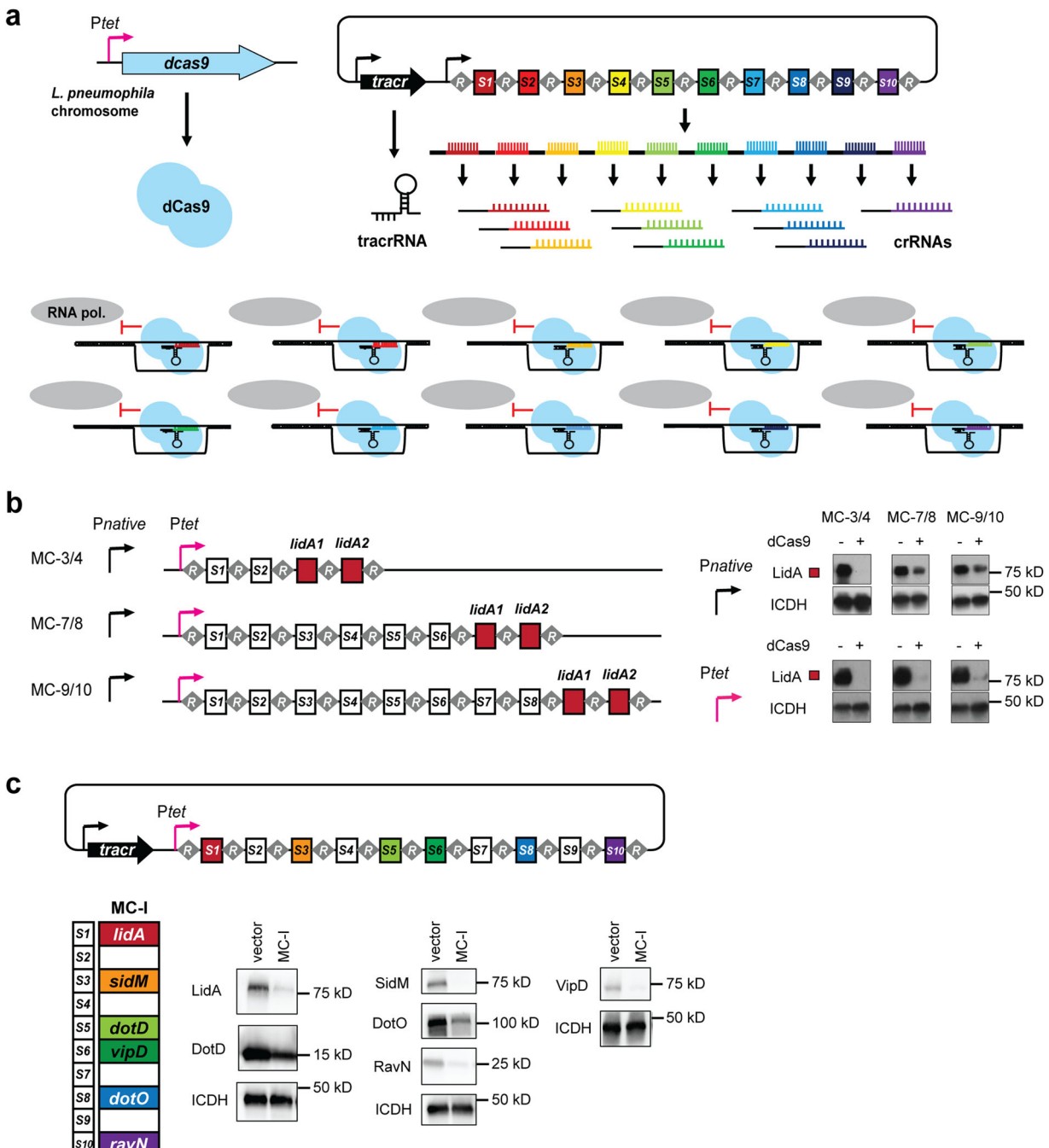

**Fig. 2 Repeat/spacer arrays facilitate multiplex gene silencing. a** Schematic representation of the theory of multiplex CRISPRi. A series of repeats, R, and spacers, S1–S10, are expressed as a single precursor-crRNA. Upon processing, individual crRNAs come together with a tracrRNA and dCas9 to simultaneously target ten unique genes for silencing. **b** Multiplex CRISPR (MC) repeat/spacer array constructs of increasing length were placed under the control of the native *S. pyogenes* promoter, $P_{native}$, or $P_{tet}$. Efficiency of *lidA* targeting by crRNAs encoded by the terminal spacers of each array was assessed by monitoring LidA protein levels in Lp02(*dcas9*) and Lp02 by immunoblot. ICDH serves as a loading control. **c** A $P_{tet}$-MC construct, MC-I, capable of expressing ten unique crRNAs was designed. crRNAs encoded by *S1*, *S3*, *S5*, *S6*, *S8*, or *S10* target genes in which antibodies directed against their encoded gene products are available. Bacteria pellets were collected from axenic cultures containing MC-I or the empty vector (+40 ng/mL aTC) and multiple immunoblots were run to accommodate the range of targets. ICDH serves as a loading control. Replicates are given in Supplementary Fig. S2.

**Spacer position influences gene silencing efficiency in $P_{tet}$-MC arrays.** Encouraged by the finding that our MC-I 10-plex construct had silenced multiple target genes, we determined how silencing efficiency varied dependent on the spacer position within the CRISPR array. To that end, we designed five additional 10-plex constructs, MC-II through MC-VI (Fig. 3a), in which the aforementioned spacers encoding crRNA$^{lidA}$, crRNA$^{vipD}$,

crRNA$^{sidM}$, crRNA$^{dotO}$, crRNA$^{dotD}$, and crRNA$^{ravN}$ were gradually shifted from proximal positions into more distal positions within the array, occupying either odd- or even-numbered positions. crRNA$^{lpg2793}$ and crRNA$^{lpg0208}$ were also encoded by spacers at various positions as these crRNAs had emerged in preliminary studies to silence very efficiently or not at all, respectively.

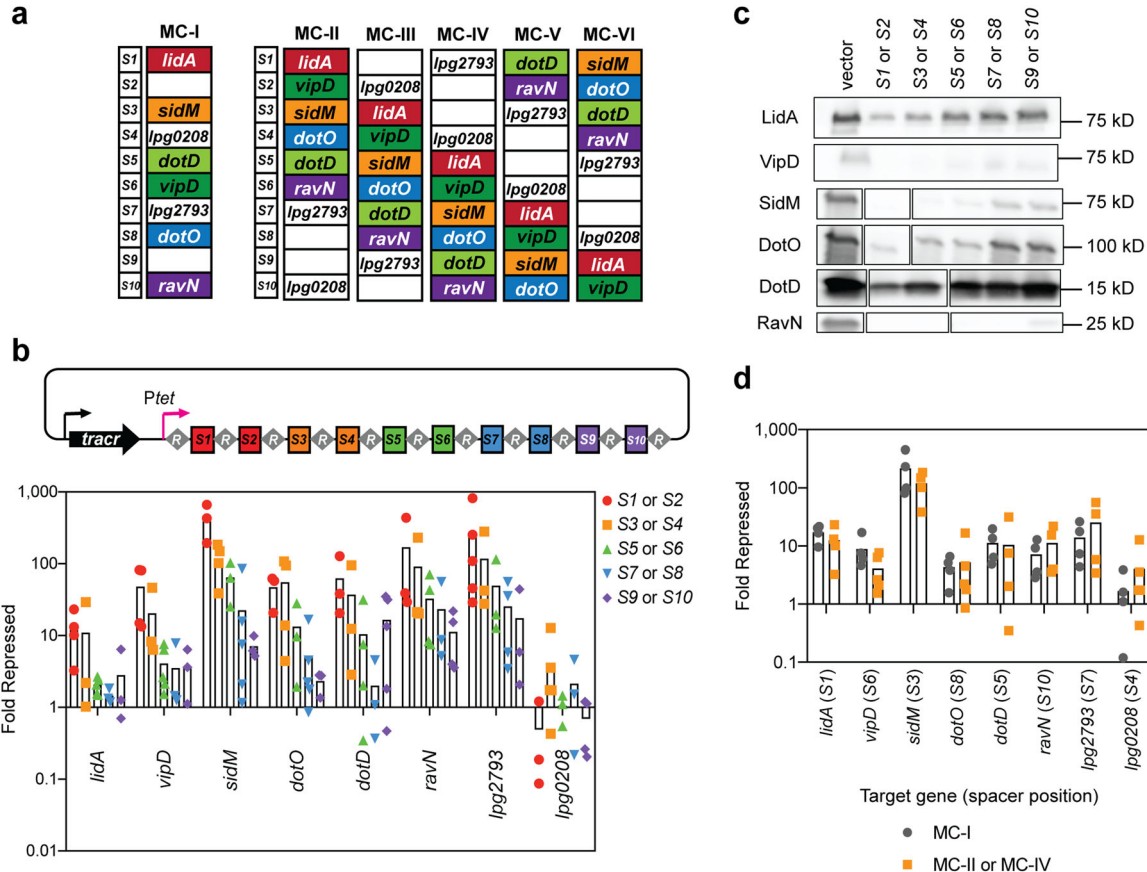

**Fig. 3 The effect of spacer position on gene silencing by P_{tet}-MC arrays. a** Additional P_{tet}-MC constructs, MC-II through MC-VI, were made to test the effect of spacer position and environment on gene silencing. Each spacer is present in either position *S1, S3, S5, S7,* and *S9* or *S2, S4, S6, S8,* and *S10*. **b** RNA was extracted from Lp02(*dcas9*) bearing either the empty vector or an MC construct after axenic culture (+40 ng/mL aTC). qPCR was used to measure mRNA levels of target genes. Fold repression was determined using $\Delta\Delta C_T$. Bars indicate the mean of multiple replicates shown as individual data points. A value of 1 indicates that the mRNA level was the same in both strains. **c** Immunoblot analyses were performed on pellets collected from the same cultures. The blot shown is a representative of three replicates, with bands rearranged corresponding to spacer position. Original immunoblots and ICDH loading controls are shown in Supplementary Fig. S3. **d** RNA was extracted from Lp02(*dcas9*) bearing either the vector or the MC-I construct after axenic culture (+40 ng/mL aTC). qPCR was used to measure mRNA levels of target genes. Fold repression was determined using $\Delta\Delta C_T$ and plotted in comparison to data first shown in **b**.

Efficiency of gene silencing by MC-II through VI after two days of induction during axenic growth was assessed for all targets at both the mRNA level and protein level and compared to that of the Lp02(*dcas9*) control strain containing the empty vector. qPCR analyses indeed revealed a polarity in gene silencing along the length of the repeat/spacer arrays, as has been noted for other CRISPRi platforms before[21,33], with spacers in the proximal and distal positions causing the most- or least-efficient gene silencing, respectively (Fig. 3b). Some crRNAs did silence their target genes very well (crRNA^{sidM}, crRNA^{ravN}, and crRNA^{lpg2793}) even when encoded from distal spacer positions. In fact, *ravN* was as efficiently silenced by crRNA^{ravN} encoded by the *S10* spacer (MC-IV) as *lidA* was by crRNA^{lidA} encoded by the *S1* spacer (Fig. 3b). crRNA^{lpg2793}, which showed continuously high levels of repression, experienced greater than an order of magnitude repression even with the crRNA encoded by spacer *S10*. Conversely, crRNA^{lpg0208} did not work well regardless of the encoding spacer position. Notably, all genes except *lpg0208* experienced at least an order of magnitude repression when targeted by crRNAs encoded from spacers at positions as distal as *S4*, and five of them showed at least an order of magnitude silencing up to spacer position *S6*. Immunoblot analysis of the six targets for which antibodies were available confirmed that protein levels correlated very well with the mRNA abundance (Fig. 3c and

Supplementary Fig. S3), with depletion being near or below the detection limit. *dotD* showed better than expected gene silencing with the most distal *S9* spacer (MC-IV) for reasons that remain to be determined.

**Gene silencing efficiency by CRISPRi is independent of spacer environment.** crRNAs within the various MCs (MC-II through MC-VI), although transcribed from spacers in different positions, were always flanked by the same pair of neighboring spacers. For example, the spacer encoding the less efficient crRNA^{lidA} was always surrounded at the 3' and 5' ends by spacers encoding crRNA^{lpg0208} and crRNA^{vipD}, respectively. Those neighboring spacers may have inadvertently affected transcription or processing of crRNA^{lidA}. To test if the spacer environment had an effect on crRNA production and therefore gene silencing efficiency, which has been proposed in the past[34], we compared target gene repression by crRNAs that were encoded by spacers at identical positions but surrounded by different spacers. crRNA^{lidA}, crRNA^{sidM}, crRNA^{dotD}, and crRNA^{lpg2793} are encoded from the same position in MC-I and MC-II (spacers *S1, S3, S5,* and *S7*), while crRNA^{lpg0208}, crRNA^{vipD}, crRNA^{dotO}, and crRNA^{ravN} originate from position *S4, S6, S8,* and *S10* in both MC-I and MC-IV, yet they are flanked by entirely different sets of spacers (Fig. 3a). Upon expression in Lp02(*dcas9*), crRNAs encoded by

MC-II or MC-IV, despite being encoded from different spacer environments, consistently achieved the same level of gene repression as those encoded by MC-I (Fig. 3d). Together, these results demonstrate that spacer position along the array has a much greater impact on multiplex CRISPRi-mediated gene silencing than the spacer environment.

**boxA elements reduce polarity of spacer transcription to promote maximum gene silencing.** While the addition of the strong $P_{tet}$ promoter did enhance gene silencing by distal spacers (Fig. 2b), gene silencing polarity within the MC arrays was still not fully overcome (Fig. 3), indicating that additional factors likely influenced crRNA production. BoxA elements are often found upstream of long non-coding RNAs, like the 16 S RNA[35], where they are bound by the Nus complex to improve RNA polymerase processivity by multiple mechanisms[36]. Recently Stringer et. al reported that boxA elements may be evolutionarily conserved upstream of naturally occurring CRISPR arrays where they appear to promote expression of downstream spacers within long arrays[37].

To investigate whether a boxA element could promote expression of our synthetic repeat/spacer array, we surveyed the levels of precursor-crRNA expression in strains bearing constructs with and without boxA in the absence of tracrRNA (to limit precursor-crRNA processing). In deciding the boxA sequence to incorporate, we compared the E. coli boxA consensus sequence with that of the Coxiella 16 S boxA and the boxA found upstream of the naturally occurring L. pneumophila sp. Lens CRISPR array (Fig. 4a). Interestingly, a boxA element was not readily identified upstream of the L. pneumophila Philadelphia-1 16 S RNA encoding region. We chose to first mutate the leader sequence of the MC-II and MC-IV constructs to include the 11 base pair L. pneumophila sp. Lens boxA sequence (5′-GTTCTTTAAAA-3′), and the five flanking base pairs on either side, at both a distance of -58 (boxA(-58)) and -90 (boxA(-90)) base pairs upstream of the first repeat while maintaining the overall length of the leader sequence (Fig. 4b, see Supplementary Fig. S4 for sequence detail). RNA extracted from axenic cultures induced with aTC was analyzed by qPCR using primer pairs that probed three different regions along the length of the anticipated precursor-crRNA. We found that boxA containing constructs do in fact express on average ~10-fold more precursor-crRNA than boxA-less strains throughout the entire length of the synthetic repeat/spacer array (Fig. 4c), suggesting that boxA does promote array transcription.

With evidence that boxA does increase precursor-crRNA levels, we next asked how gene silencing was affected. Upon introducing boxA in either the -58 or -90 position for all MC constructs (now with tracrRNA present to promote crRNA processing into individual gene targeting units), mRNA and protein analysis of construct-bearing Lp02(dcas9) strains revealed that this boxA element greatly resolved gene silencing polarity for the majority of the target genes tested (Fig. 4d and Supplementary Fig. S5). The presence of boxA in either the -58 or -90 position maximized silencing of sidM, dotO, dotD, ravN, and lpg2793 by crRNAs encoded from all spacer positions, even S9 or S10, presumably by allowing transcription of the repeat/spacer array to proceed throughout the entire length of the array. sidM, dotD, ravN, and lpg2793 displayed nearly two orders of magnitude gene repression from crRNAs encoded from all spacer positions, a vast improvement over the diminishing repression observed by boxA-less constructs. In all cases, the degree of mRNA depletion was mirrored by the reduction in protein levels. Minor discrepancies in dotD and DotD levels suggest slower protein turnover rates/increased protein stability of DotD compared to

other proteins probed. Silencing of lidA, lpg0208, and vipD was not altered by the addition of boxA, suggesting that other factors such as intrinsic gene regulation and genomic environment were contributing to the expression of the target genes and were possibly inhibiting silencing efficiency. In conclusion, the processivity of transcription of long repeat/spacer arrays is the major driving factor of achieving multiplex gene silencing even from distal spacer positions.

**CRISPRi is functional during L. pneumophila intracellular growth.** While CRISPRi has been performed in a variety of bacteria during growth in axenic culture, gene silencing had yet to be accomplished in any pathogen during infection. We confirmed that our platform was functional in L. pneumophila undergoing intracellular growth and that, upon repression of specific targets, we could reproduce previously reported intracellular growth phenotypes.

L. pneumophila requires the Dot/Icm T4SS to deliver effector proteins into the host cell in order to establish a replication vacuole. Strains that bear loss-of-function mutations in the genes encoding T4SS subunits are unable to establish this specialized vacuole and fail to replicate[38]. The ATPase DotO and the outer membrane protein DotD are components critical for the function of the T4SS transporter, and L. pneumophila mutants with disruptions in their encoding genes (lpg0456 and lpg2674, respectively) are attenuated for virulence[39–41]. Plasmids encoding crRNAs directed against dotO (crRNA$^{dotO}$) or dotD (crRNA$^{dotD}$) were individually introduced to Lp02(dcas9) and, after 48 h of CRISPRi, decreased protein levels of DotO and DotD were verified by immunoblot analysis (Fig. 5a). It is important to note, and well established[39,40], that mutations disabling the T4SS have no influence on the general fitness of L. pneumophila outside the context of the host. Human-derived U937 macrophages[42,43] were then challenged with these crRNA-expressing strains, and bacterial growth was monitored over a period of 72 h (Fig. 5a). While the Lp02(dcas9) strain containing the empty plasmid grew several orders of magnitude, the CRISPRi strains depleted of either DotO or DotD were impaired for growth at a level comparable to that of the avirulent mutant strain Lp03 which bears a chromosomal mutation in the dotA gene that encodes another critical subunit of the T4SS[27].

In agreement with this result, our efforts to target and repress an effector-encoding gene also proved successful. MavN is a multi-pass membrane protein from L. pneumophila that is essential for delivering metal ions across the LCV membrane into the vacuolar lumen[44,45]. Deletion of mavN (lpg2815) from the L. pneumophila chromosome results in a complete growth defect in A/J mouse macrophages[44]. Upon targeting mavN with crRNA$^{mavN}$, we observed a similarly dramatic growth defect in U937 macrophages in the Lp02(dcas9) strain, while the Lp02 control strain grew robustly over the same time period (Fig. 5b). In the absence of an antibody specific to MavN, mRNA levels were analyzed by qPCR and confirmed mavN to be repressed more than ten-fold in the Lp02(dcas9) strain relative to the Lp02 control strain. These data indicate that CRISPRi is functional in L. pneumophila both during growth in axenic media and during a multi-day infection experiment.

**Multiplex CRISPRi reveals gene dosage effects during intracellular growth.** Next, we explored if intracellular growth of L. pneumophila can be manipulated through multiplex CRISPRi. We created dotO-specific $P_{tet}$-MC constructs (called MC-II-dotO to MC-VI-dotO; Fig. 6a) that mimicked $P_{tet}$-MC-II through MC-VI, except that all crRNA-encoding spacers besides the crRNA$^{dotO}$-encoding spacer were scrambled, rendering them incapable

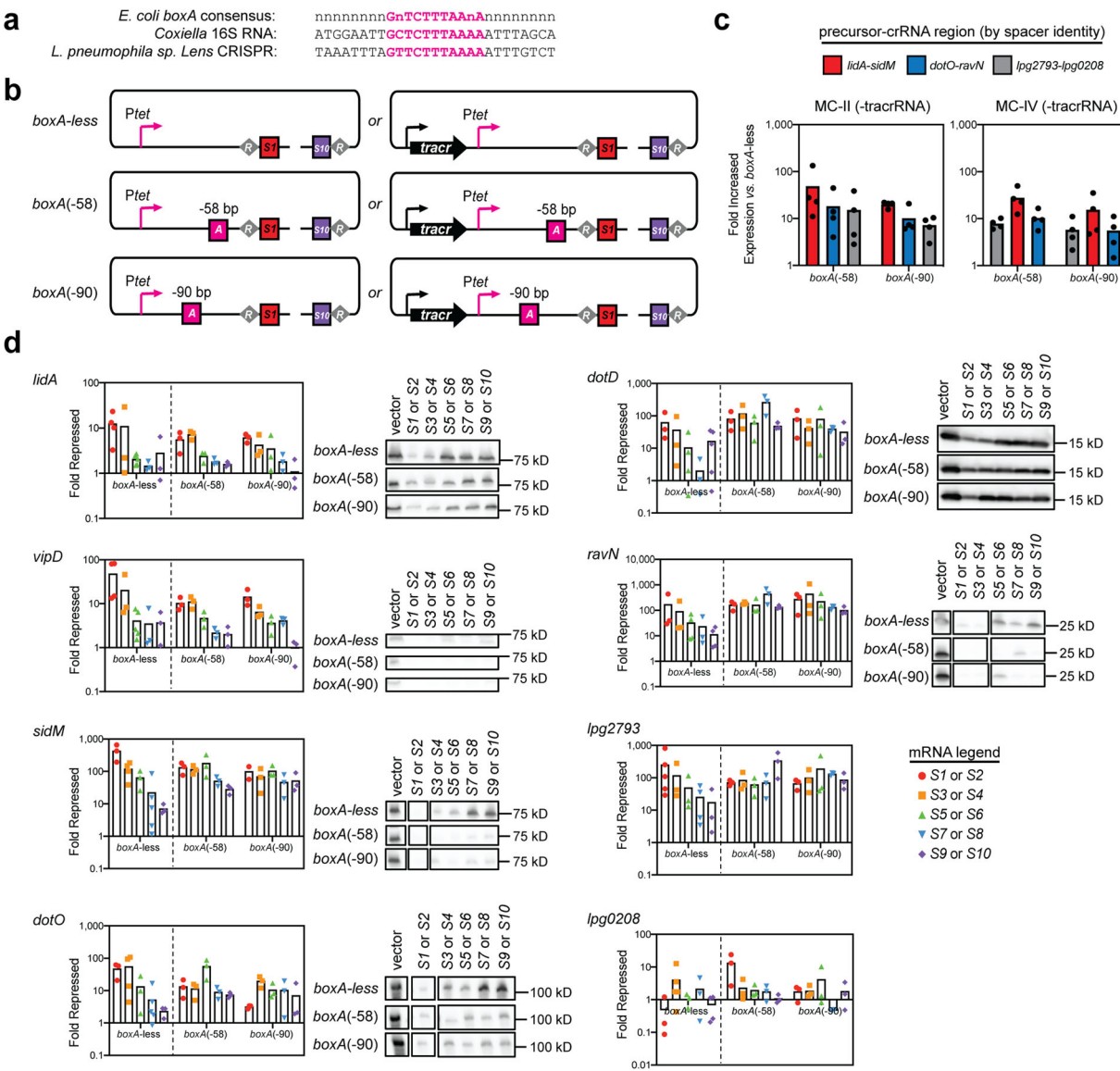

**Fig. 4 The effect of *boxA* elements on gene silencing by P*tet*-MC arrays. a** Sequence alignment of the *boxA* elements identified in *E. coli* (*boxA* consensus), *Coxiella* (16 S RNA encoding sequence), and *L. pneumophila sp. Lens* (CRISPR array). **b** The *boxA* motif from the *L. pneumophila sp. Lens* CRISPR array was added to the leader sequence of the P*tet*-MC constructs either -58 (*boxA*(-58)) or -90 (*boxA*(-90)) base pairs upstream of the repeat/spacer array. *boxA* is depicted as a pink square. **c** Constructs lacking the tracrRNA-encoding sequence were used to probe precursor-crRNA expression from *boxA*-containing *vs.* *boxA*-less constructs in axenic cultures of Lp02(*dcas9*) (+40 ng/mL aTC). Precursor-crRNA levels were measured by qPCR using primer pairs that annealed to three different regions of the precursor-crRNA. Fold increased expression *vs.* *boxA*-less constructs was determined using ΔΔC$_T$. Data are ordered by spacer position in constructs MC-II or MC-IV. **d** Constructs with the tracrRNA-encoding sequence were used to probe gene repression by qPCR of RNA from axenic cultures of Lp02(*dcas9*) bearing *boxA* constructs to that with the empty vector (+40 ng/mL aTC). Fold repression was determined using ΔΔC$_T$. For comparison, *boxA*-less qPCR data from Fig. 3 is shown again. For visualization of protein levels, pellets from *boxA*-less and *boxA*(-58) and *boxA*(-90) strains were collected and processed for immunoblot analysis. The bands were reordered corresponding to their spacer position. Original immunoblots and ICDH loading controls are shown in Supplementary Fig. S5.

of base pairing with their original targets. Notably, the base pair count and composition ("agtc" content) of the scrambled spacers were maintained, and all repeat sequences remained intact. As a result, these multiplex arrays should encode only crRNA$^{dotO}$ as functional crRNA from spacer positions S2, S4, S6, S8, or S10 for direct comparison to the single crRNA$^{dotO}$-encoding CRISPRi construct (from Fig. 5). Furthermore, we created a control construct, MC-VII-dotO, in which the *dotO*-targeting spacer in position S2 was scrambled to provide evidence that any observed phenotypes were a direct result of silencing *dotO*.

We examined *dotO* repression by the new constructs in axenic culture to establish the degree of *dotO* silencing efficiency. qPCR

revealed that *dotO* silencing by MC-VI-dotO and MC-II-dotO constructs with crRNA$^{dotO}$-encoding spacers in the proximal S2 and S4 positions, respectively, was robust and comparable to that of the single crRNA$^{dotO}$-encoding CRISPR construct (Fig. 6b). *dotO* silencing efficiency by the MC-III-dotO to MC-V-dotO constructs gradually decreased with increasing distal shift of the crRNA$^{dotO}$-encoding spacer in these arrays. As expected, the scrambled *dotO* spacer in MC-VII-dotO did not promote gene silencing, showing that the effects of MC-II-dotO to MC-VI-dotO was specific to crRNA$^{dotO}$. Again, immunoblot analyses of cell pellets collected from these axenic *L. pneumophila* cultures showed depletion of DotO to the same extent as would be

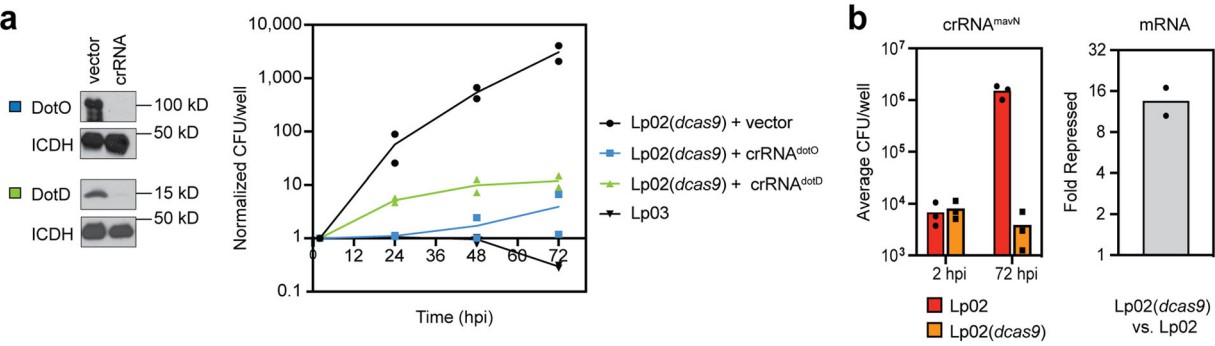

**Fig. 5 _L. pneumophila_ CRISPRi is functional during intracellular growth. a** Immunoblots confirm decreased levels of DotO and DotD in Lp02(_dcas9_) strains expressing crRNA^dotO or crRNA^dotD. ICDH serves as a loading control. Growth of these strains in U937 macrophages was compared to that of Lp02 (_dcas9_) bearing an empty vector and the avirulent strain Lp03, with a chromosomal mutation in _dotA_. Growth is plotted as CFU/well normalized to the CFU at 2 h post infection (hpi). **b** Growth of Lp02 and Lp02(_dcas9_) expressing crRNA^mavN was assayed in U937 macrophages. Growth is given as the average of CFU/well of three experiments at 2 and 72 hpi. mRNA levels of _mavN_ in Lp02(_dcas9_) _vs._ Lp02 were measured by qPCR. Fold repression was determined using $\Delta\Delta C_T$.

predicted from the mRNA abundance data (Fig. 6c and Supplementary Fig. S6).

We then challenged U937 macrophages with these _boxA_-less crRNA^dotO-expressing strains and compared their growth over a period of 48 h to that of Lp02(_dcas9_) with the empty vector, which served as a measure of maximum growth, and Lp03, which served as a measure of maximum attenuation (Fig. 6d, top). Strikingly, the severity of the growth defect precisely mirrored the reduction in _dotO_ mRNA levels caused by these constructs, with the most severe defect caused by crRNA expression from the _S2_ spacer (MC-VI-dotO construct) and lesser defects observed with the distal shift of the _dotO_ spacer in MC-II-dotO to MC-V-dotO. The scrambled _dotO_ spacer of MC-VII-dotO was unable to silence _dotO_ and allowed proficient _L. pneumophila_ growth comparable to the Lp02 control strain. Thus, the polarity of spacer expression and the corresponding decay of target repression represents a titrated gene silencing platform.

Since silencing was least efficient for the MC-V-dotO construct (_S10_, purple triangle in Fig. 6d, top), we added a _boxA_(-58) motif to the leader sequence of the array and re-examined the effect of this _boxA_(-58)-MC-V-dotO construct on _L. pneumophila_ intracellular growth. Remarkably, we now observed a significant attenuation of growth in U937 macrophages over 48 h, achieving over an order of magnitude replication defect compared to Lp02 (_dcas9_) bearing the empty vector (Fig. 6d bottom, _p_ = 0.033, one-unpaired _t_-test). In fact, with the _boxA_(-58) motif present, the growth defect observed by expressing the crRNA from the _S10_ spacer was now comparable to that of expression from the _S6_ spacer in a _boxA_-less construct (green diamond in Fig. 6d, top). mRNA and protein analyses confirmed that _dotO_ silencing by the _boxA_(-58)-MC-V-dotO construct mirrored that of the _dotO_ silencing in the _boxA_-less MC-III-dotO construct (Figs. 6b and 6c). Together, these data demonstrate that multiplex gene silencing by long CRISPR arrays can be efficiently boosted even within infection models simply by positioning a _boxA_ element in the leader region.

**_boxA_-MC constructs allow for multi-gene knockdown throughout infection**. At last, we tested if our multiplex CRISPRi platform, bearing both the $P_{tet}$ promoter and a _boxA_(-58) element, is capable of targeting multiple _L. pneumophila_ genes simultaneously throughout the course of host infection. To do so, we constructed a _boxA_(-58) MC 10-plex construct with ten unique spacers targeting ten unique predicted effector-encoding genes (Fig. 7a; targets in order: _lpg2300, lpg1129, lpg1689, lpg0621_,

_lpg1663, lpg2128, lpg0921, lpg2344, lpg2793, lpg0902_). Targets were chosen at random and, while their protein products have not yet been characterized in detail, at least eight of the ten have been experimentally proven to be a translocated cargo of the T4SS[46–48]. Consistent with the hypothesis of high redundancy amongst _L. pneumophila_ effectors, we did not observe differences in intracellular growth between Lp02(_dcas9_) bearing either the empty vector or _boxA_(-58) MC 10-plex upon challenging U937 macrophages for 48 h (Fig. 7b). Equal numbers of bacteria were subsequently isolated from U937 macrophages and qPCR analysis of extracted RNA revealed that multiplex CRISPRi mediated repression of the majority of targets after 48 h of infection. In fact, eight of the ten targets were repressed nearly an order of magnitude or more (Fig. 7c). These data suggest that _boxA_ multiplex CRISPRi is a robust, multi-gene silencing technique even during host infection.

## Discussion

In this study, we have established a multiplex CRISPRi platform in the pathogen _L. pneumophila_ and provide proof-of-concept for this platform to be usable not only during growth in axenic media but also during macrophage infection where it reproduced known intracellular growth phenotypes (Fig. 6). Importantly, by placing the crRNA-encoding spacer in positions further downstream within the array, the degree of gene silencing was titratable (Fig. 3). In contrast, when combined with a _boxA_ element, our 10-plex CRISPR array had the potential to silence up to ten unique genes simultaneously (Fig. 4 and Fig. 7) making it a powerful tool to study even synergistic genetic interactions.

Our methodical approach to analyze the potential of multiplex CRISPRi not only allowed us to generate a powerful platform capable of silencing up to ten unique genes, a platform that may prove invaluable for studying the more than 300 _L. pneumophila_ effectors in the future, but it also revealed additional surprises that could be applied to future gene interrogations. For example, some of our MC arrays encoded not just one but two crRNAs (called _lidA1_ and _lidA2_) towards the same target gene (Fig. 2b), resulting in LidA to be efficiently depleted even though the encoding spacers were located in the most distal spacer positions (_S9_ and _S10_) within the array. Notably, when only the crRNA encoded from the _lidA1_ spacer was used in subsequent MC constructs (Fig. 3), silencing was not nearly as efficient, suggesting that the combined action of both spacers in $P_{tet}$-MC-9/10 promoted maximum silencing. While only a single example, future users of this technology

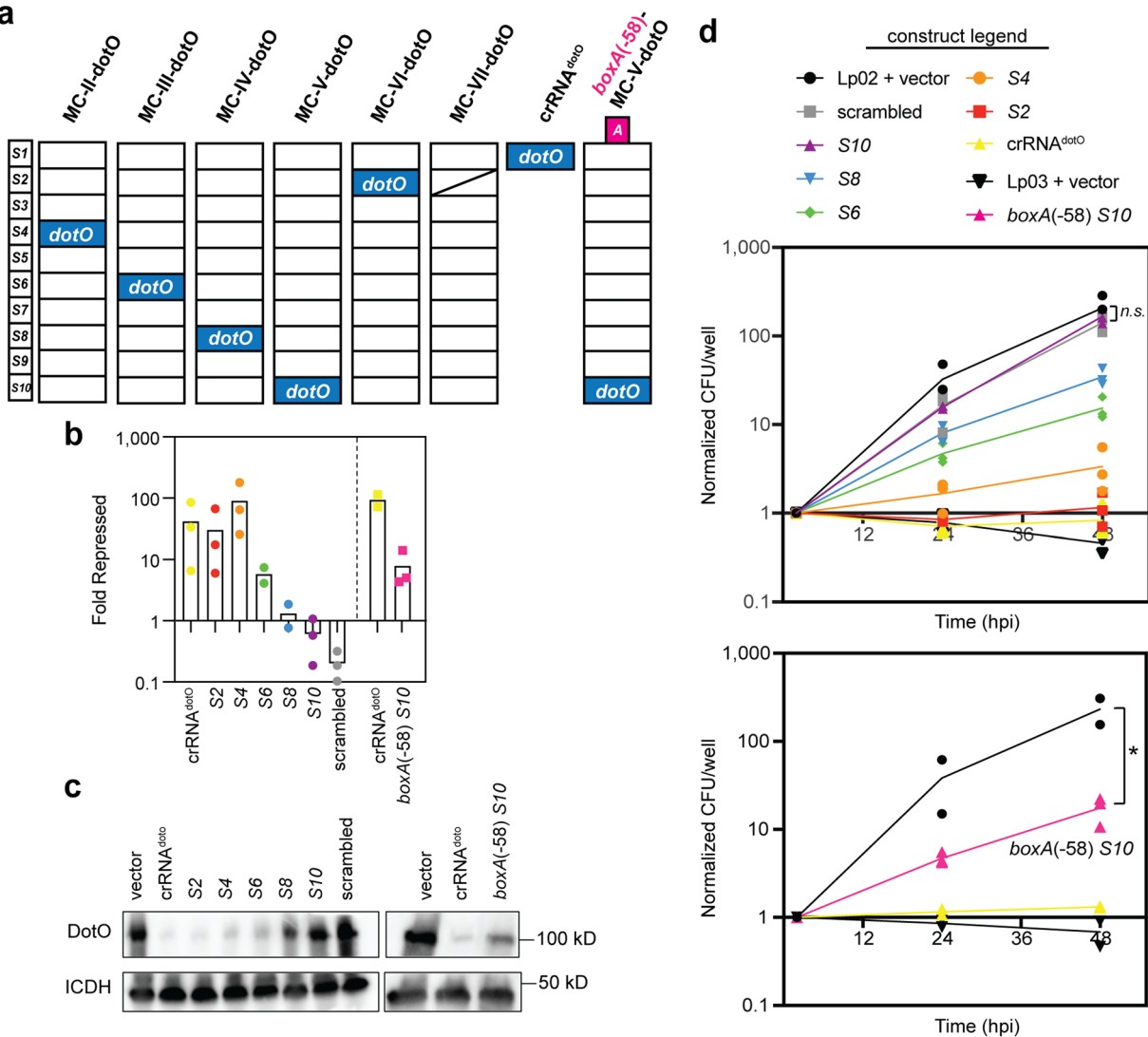

**Fig. 6 Gene repression during infection is titratable with multiplex CRISPRi. a** *boxA*-less, *dotO*-targeting P$_{tet}$-MC constructs, MC-II-dotO, MC-III-dotO, MC-IV-dotO, MC-V-dotO, and MC-VI-dotO, were constructed such that the sequences of all spacers, other than *dotO*, were scrambled. The *dotO* spacer can be found in the same position as in the original MC-II through MC-VI constructs. MC-VII-dotO is identical to MC-VI-dotO except that the *dotO* spacer is also scrambled. crRNA$^{dotO}$ is the same as in Fig. 5. Later, a *boxA* sequence was added in the -58 position to MC-V-dotO to create *boxA*(-58) *S10*. **b** RNA extracted from axenic cultures of Lp02(*dcas9*) bearing these constructs was compared to that of Lp02(*dcas9*) bearing the empty vector by qPCR, as in Fig. 3 (+40 ng/mL aTC). **c** Immunoblot analyses were performed on pellets collected from the same cultures and samples were run according to *dotO* spacer position. Immunoblot replicates are found in Supplementary Fig. S6. ICDH serves as a loading control. **d** aTC-induced Lp02(*dcas9*) bearing *boxA*-less and *boxA*(-58) constructs were used to infect U937 macrophages. Growth of these strains and Lp02(*dcas9*) and Lp03 bearing the empty vector was monitored over 48 h post infection (hpi). Colony forming unit (CFU) counts were normalized to the count at two hours post infection. Normalized counts for each experiment at each timepoint are shown. *n.s.* = not significant ($p = 0.44$, $n = 3$), *$p = 0.033$, $n = 2$ or 3, one-unpaired *t*-test. All replicates are from distinct experiments.

could experiment by adding more than one spacer for a given gene to enhance gene silencing. crRNAs encoded by spacers *lidA1* and *lidA2* target different regions of *lidA*, but one could hypothesize repeating the same spacer sequence could also increase gene silencing of a target by producing more of the crRNA. The malleability of our MC scaffold allows users who may not need to target ten unique genes to apply these tactics to achieve maximum silencing levels of fewer gene targets.

Our platform also provides two different strategies for titratable gene silencing. First, since both *dcas9* and the CRISPR array are under the regulation of the P$_{tet}$ promoter, simply altering the amount of aTC inducer presumably changes gene silencing by controlling both the quantity of dCas9 and the abundance of the crRNA (Fig. 1b). While

regulating crRNA expression with a variety of inducers (e.g. aTC, arabinose, xylose, and IPTG) is common place, gene silencing in this way has been argued to be noisy[49]. Second, titrated gene silencing was also accomplished when P$_{tet}$-MC constructs without a *boxA* motif were used that were prone to diminishing transcription (Figs. 3 and 6). DotO constructs provided a clear example of how gene silencing through placing spacers at ever distant positions within the array can lead to titrated effects to the biological system (Fig. 6d). Recent efforts have been made to tailor the degree of crRNA-targeting of a gene by creating numerous variations of the spacer sequence away from the perfect match[49,50]. We imagine shifting the ideal spacer downward in the array would be a much simpler feat.

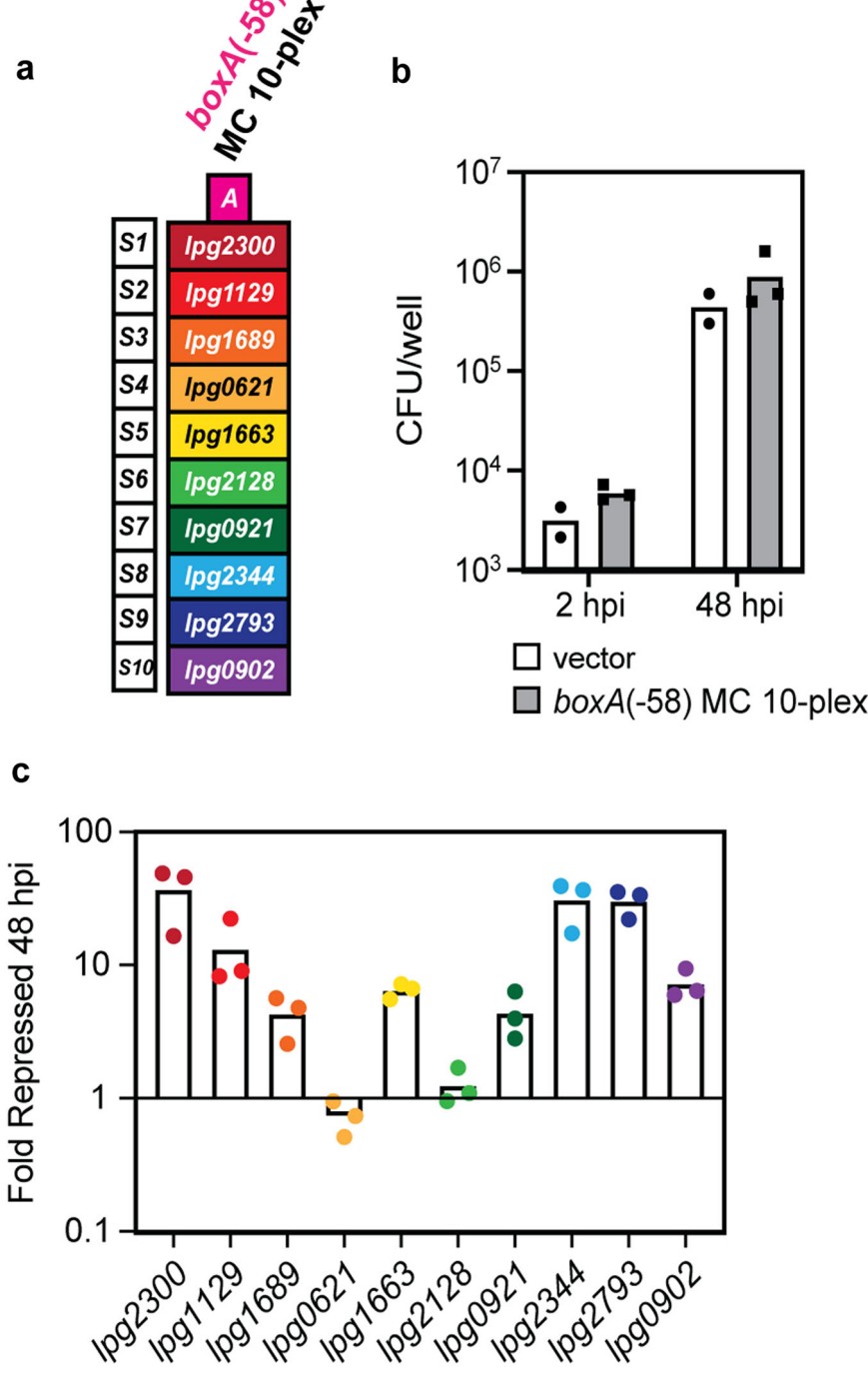

**Fig. 7 Multi-gene knockdown by *boxA*-MC persists throughout infection. a** *boxA*(-58) MC 10-plex was constructed with ten unique spacers for expression of crRNAs targeting ten unique predicted effector-encoding genes. **b** Lp02(*dcas9*) bearing either the empty vector or *boxA*(-58) MC 10-plex was used to infect U937 macrophages, and intracellular growth was determined by measuring the average of CFU/well of two to three experiments at 2 and 48 h post infection (hpi). **c** At 48 hpi bacteria were extracted from the U937 macrophages and harvested for RNA. mRNA levels of each of the ten target genes was measured by qPCR. Fold repression of target genes in *boxA*(-58) MC 10-plex compared to vector-bearing Lp02(*dcas9*) was determined using $\Delta\Delta C_T$.

Using CRISPRi in *L. pneumophila* allowed for the systematic investigation of not only repeat/spacer arrays as a gene targeting platform, but also provided additional insight into how CRISPR arrays might function in nature. We show that with the native *S. pyogenes* promoter, only the most proximal spacers are transcribed (Fig. 2b). It is known that spacers acquired from most recent infections are inserted at the first position in the array[51]. Our data reiterate that this is an advantageous strategy as the

earliest spacers in the array are the most likely to be transcribed, and therefore, will produce the greatest quantity of protective crRNAs for an ongoing viral attack. Several bacterial CRISPR systems appear to have overcome polarity of spacer expression through acquiring *boxA* sequences upstream of their arrays[37]. *boxA* could improve RNA polymerase processivity of these and our synthetic arrays (Fig. 4) by a number of different mechanisms, one of which may be blocking Rho-dependent transcription

termination[52–54], as hypothesized by Stringer et al.[37]. Still, as naturally occurring CRISPR arrays can stretch to the hundreds of spacers[32], there could be additional unknown factors that regulate spacer transcription that could one day be applied to a CRISPR-based tool.

Ultimately, our *dotO* (Fig. 6) and MC 10-plex (Fig. 7) gene silencing experiments during host infection provided evidence that this multiplex CRISPRi technology is ready to be applied to biological investigations, especially in the study of *L. pneumophila* pathogenesis. As touched on above, *dotO* expression appears to be exquisitely tailored towards promoting maximum infection, as any deviation of expression led to growth attenuation in macrophages (Fig. 6). In what could be a nutrient limited environment, it seems *L. pneumophila* expresses just enough DotO in nature to achieve virulence through T4SS assembly and secretion of effectors, without being wasteful by producing an excess amount of this protein.

The *Legionella* community has long thought that the multitude of *L. pneumophila* effectors combined with the lack of growth phenotypes observed upon creating single deletion strains supports the existence of redundancy and synergy amongst the effectors[55]. Looking forward, the multiplex CRISPRi approach developed here holds the promise of one day probing functional overlap amongst the hundreds of *L. pneumophila* effectors. Not only can genes be silenced in bulk groups during infection (MC 10-plex, Fig. 7), but the mobility of our single plasmid based CRISPRi platform allows for easy transfer of MC constructs into a variety of *Legionella* mutant strain backgrounds to directly assess redundancy, presuming they have been equipped with a copy of *dcas9*. These analyses are not limited to macrophage infections, but any number of hosts. In this capacity, multiplex CRISPRi serves as an initial discovery tool that provides guidance on which proteins to focus on during future analyses. As a general rule, phenotypes observed using knockdown technologies such as this should always be followed-up by construction of true deletion strains to alleviate the possibility of off-target effects or polar effects related to genomic context (e.g., operons). Finally, when adapted for use in other microbial pathogens, the multiplex CRISPRi technology developed here has the potential to promote understanding of their biology, and possibly even foster the discovery of prospective drug targets.

## Methods

**Construction of *dcas9*-containing strains.** *dcas9* was added to the chromosome of *Legionella pneumophila* Philadelphia-1 Lp02 (*thyA hsdR rpsL*) through allelic exchange to make MML109 (Lp02(*dcas9*)). The tet[R]-*dcas9* segment of pdCas9-bacteria (Addgene #44249) was amplified with BKMP108-109 and introduced into pDonorP4r-P3r (Invitrogen) to generate pMME1080. The N- and C-terminus of *thyA* (*lpg2868*) were amplified by BKMP94-95 and BKMP98-99 primers and introduced into pDonorP1-P4 and pDonorP3-P2 (Invitrogen) to generate pMME1084 and pMME1094, respectively. These three donor vectors were introduced into pNPTS138_Cm-DEST (pMME1020, constructed by placing the Gateway DEST sequence on pNPTS138_Cm (Addgene #41891) at HindIII and SpeI) by a Gateway LR reaction (Invitrogen) to generate pMME1115 (N-terminus *thyA*::tet[R]-*dcas9* cassette::C-terminus *thyA*). pMME1115 was introduced into Lp02 by electroporation and strains containing the plasmid were selected for on CYET-Chloramphenicol (Cm) plates (CYET plates[56]). Cm-resistant colonies were patched on CYET plates containing 5% sucrose to remove the plasmid backbone, leaving behind the N-terminus *thyA*::tet[R]-*dcas9* cassette::C-terminus *thyA* by homologous recombination. *dcas9* incorporation into *Legionella* chromosome was confirmed by immunoblot analyses and whole genome sequencing. Primer sequences are listed in Supplementary Data 3.

**Construction of single CRISPRi constructs and preliminary multiplex CRISPRi arrays.** crRNA-expressing constructs were built using Invitrogen Gateway-compatible plasmids as depicted in Supplementary Fig. S1a. The BsaI-CRISPR segment of pCRISPR (Addgene #42875) was amplified with BKMP185-186 and introduced into pDonorP5-P2 (Invitrogen) to generate pMME1540. Specific single crRNA-encoding spacer sequences were designed as explained in Fig. 1 and added to the BsaI-CRISPR segment of pMME1540 as previously described for

pCRISPR[29]. To make preliminary multiplex CRISPRi constructs as shown in Fig. 2b, 21 crRNA-encoding spacers separated by repeats were synthesized by GenScript and provided on a pUC57 vector (pMME1170). CRISPR backbones containing four (MC-3/4), eight (MC-7/8), or ten (MC-9/10) crRNA-encoding spacers were amplified from this plasmid using forward primers crsidF_F, crsidD_F, crsetA_F, respectively, and reverse primer crlidA2_R. PCR products were BsaI treated and ligated into pMME1540 (P*native*) or pMME1748 (P*tet*) as above.

Next, the *Streptococcus pyogenes* tracrRNA-encoding sequence from pCas9 (Addgene #42876) was amplified with BKMP45-46 and introduced into pDonorP1-P5 (Invitrogen) to generate pMME985. The crRNA donor plasmids, for both the single CRISPRi constructs and the preliminary multiplex CRISPRi arrays, and pMME985 were introduced into pMME977 by the Gateway LR reaction to generate the final crRNA-encoding plasmids. Final plasmids were introduced to Lp02(-*dcas9*) or Lp02(*dcas9*) by electroporation and the strains containing the plasmid were selected for on CYE media. All final strains and crRNA-encoding spacer sequences are listed in Supplementary Data 2. Primer sequences are listed in Supplementary Data 3.

**Construction of P*tet*-MC multiplex CRISPRi constructs.** Multiplex crRNA-expressing constructs were built from GenScript-synthesized plasmids as depicted in Supplementary Fig. S1b and the nucleotide scaffold for P*tet*-MC is given in Supplementary Fig. S7a. MC-I through MC-VI, MC-II-dotO through MC-VII-dotO, and MC 10-plex array sequences were provided on a pUC57 vector (Supplementary Data 1). Next, the MC array within pUC57 was amplified using Gateway5-Ptet and Gateway2-T1term and introduced into pDonorP5-P2 by the Gateway BP reaction to generate donor plasmids. The MC backbone vector, containing simply the leader sequence of the multiplex CRISPRi array, was amplified from a GenScript plasmid using Gateway5-leader and Gateway2-leader and similarly introduced into pDonorP5-P2. These donor plasmids and pMME985 were introduced into pMME977 by the Gateway LR reaction to generate the multiplex CRISPRi constructs listed in Supplementary Data 2. These plasmids were then introduced to Lp02(*dcas9*) by electroporation and the strains containing the plasmid were selected for on CYE plates without thymidine. All final strains and crRNA-encoding spacer sequences are listed in Supplementary Data 2. Primer sequences are listed in Supplementary Data 3.

**Addition of *boxA* to P*tet*-MC multiplex CRISPRi constructs.** The *boxA* sequence from *L. pneumophila sp. Lens* was introduced to the leader region of the MC-II through MC-VI, MC-V-dotO, and MC 10-plex arrays in our donor plasmids by quickchange PCR with Pfu Turbo Polymerase (Agilent #600250-52). *boxA*(-58) was added using primers BoxA_srtF/R and *boxA*(-90) was added using primers BoxA_farF/R. Then sequence-verified donor plasmids and either pMME985 (+tracrRNA) or pMME2603 (-tracrRNA; made by adding the leader sequence of a MC construct to pDonorP1-P5 with attB1-lessTR/ attB5r-lessTR) were introduced into pMME977 by the Gateway LR reaction as before. Again, these plasmids were introduced to Lp02(*dcas9*) by electroporation and the strains containing the plasmid were selected for on CYE plates. The nucleotide scaffold for *boxA*(-58) P*tet*-MC is given in Supplementary Fig. S7b. All final strains and crRNA-encoding spacer sequences are listed in Supplementary Data 2. Primer sequences are listed in Supplementary Data 3.

**Axenic growth of *L. pneumophila* CRISPRi strains.** For experiments with single CRISPRi constructs (Figs. 1 and 5), *L. pneumophila* were grown overnight in AYE (10 g ACES, 10 g yeast extract per liter, pH 6.9 with 0.4 mg/ml cysteine and 0.135 mg/ml ferric nitrate) under inducing conditions (by adding either 20 ng/mL or 40 ng/mL anhydrous tetracycline (aTC, Clontech #631310)). For all other experiments, *L. pneumophila* cultures were grown overnight in AYE under non-inducing conditions (-aTC). On the second day, cultures were sub-cultured twice (AM and PM, ~6–7 h apart) to OD$_{600}$ 0.2–0.3 with 2–3 mL fresh AYE containing 40 ng/mL aTC. On the third day, cultures that had reached OD$_{600}$ 3–5 (post-exponential growth) were collected for mRNA analyses, immunoblot analyses, and/or use in host cell infections.

**Immunoblot assays and antibodies.** Immunoblot assays were performed on bacteria pellets resuspended in SDS sample buffer to either $1\times10^8$ CFU or OD$_{600}$ = 10. SDS-PAGE gels were run on a Protein III system and transfers were performed with the Trans-Blot Turbo Transfer System (BioRad). Protein bands were detected using a primary antibody followed by an HRP-conjugated secondary antibody (rabbit, Life technologies #G21234, or mouse, Invitrogen #G21040) and visualized via chemiluminescence using Clarity Western ECL Substrate and the ChemiDoc MP Imaging System (BioRad). Primary antibodies directed against SidM[57], VipD[58], RavN[59], and LidA[60] were described before. Primary antibody against ICDH was a kind gift of Abraham (Linc) Sonenshein (Tufts University School of Medicine). Primary antibodies against DotD and DotO were a kind gift of Joseph B. Vogel (Washington University in St. Louis). Anti-Cas9 monoclonal antibody was purchased from Active motif (Catalog #61577). All uncropped immunoblots are found in Supplementary Fig. S8.

**RNA extraction and qPCR.** Bacterial pellets were obtained from liquid culture or harvested from host cells. Bacteria from macrophages were harvested using digitonin extraction, as described below for intracellular growth assays, and pelleted at 10,000$g$ for 10 min. Bacterial RNA extraction was performed on bacteria pellets using the Trizol Max Bacterial RNA Isolation Kit (Invitrogen #16096040). Contaminating DNA was removed using the Turbo DNA-free Kit (Invitrogen #AM1907) and RNA was converted to cDNA using the High-Capacity cDNA Reverse Transcription Kit (Applied Biosystems #4368814). qPCR was performed using the SYBR Green Master Mix (Applied Biosystems #4367659) on a StepOnePlus Real-Time PCR System (Applied Biosystems) using comparative $C_T$ and the standard 2-hour protocol. qPCR primers (found in Supplementary Data 3) were designed using NCBI Primer-BLAST such that they amplified ~100–150 bp of sequence near the 5' end of the gene of interest and had a melting temperature between 57 °C and 63 °C. mRNA levels from different samples were normalized to the house-keeping gene ($rpsL$) levels and mRNA levels in CRISPRi strains were compared to that of -dCas9 strains or the empty vector control strain using the $\Delta\Delta C_T$ method[61] to determine fold repression. Similarly, mRNA levels in $boxA$-bearing MC CRISPRi strains (-tracrRNA) were compared to that of $boxA$-less MC CRISPRi strains (-tracrRNA) using the $\Delta\Delta C_T$ method to determine increased precursor-crRNA expression.

***L. pneumophila* intracellular growth assay in human derived U937 cells.** U937 monocytes (ATCC CRL-1593.2) were maintained in DMEM + 10% FBS + glutamine. Three days prior to challenge with *L. pneumophila*, cells were plated on 24-well plates at $3\times10^5$ cells/well with 0.1 μg/mL 12-O-tetradecanoylphorbol-13-acetate (TPA, Sigma-Aldrich #P1585) to promote differentiation. *L. pneumophila* strains were incubated under inducing conditions, as described above, and added to differentiated U937 macrophages in DMEM + 10% FBS + glutamine containing 40 ng/mL aTC at a multiplicity of infection (MOI) of 0.05. Plates were centrifuged for five minutes at 200 x $g$ to increase cell-cell contact. After a 2-h incubation, extracellular bacteria were removed by washing cells twice with DMEM + FBS + glutamine media containing 40 ng/mL aTC. Bacteria were collected 2, 24, 48, or 72 h post infection (hpi). To extract the bacteria from the macrophages, digitonin (0.02% final concentration) was added to each well and incubated 10 min at 37 °C. Subsequently, lysate was collected, and each well was rinsed with dH$_2$O to ensure collection of all bacteria. Bacterial samples were serially diluted and spotted on CYE plates to determine CFU. Results are given as the CFU normalized to the CFU at 2 hpi.

**Statistics and reproducibility.** Statistical comparison of growth between MC-V-dotO and Lp02($dcas9$) bearing the empty vector or $boxA$(-58)-MC-V-dotO and Lp02($dcas9$) bearing the empty vector in Fig. 6 was carried out in GraphPad Prism 8 using the $t$-test analysis function. A one-unpaired $t$-test was performed without correction for multiple comparisons and without assuming a consistent SD. Changes in growth were considered significant if $p < 0.05$. Two to three replicates were obtained for each strain and replicates were obtained from distinct experiments. Replicate data for all other experiments of this manuscript are shown as individual data points and were similarly collected from distinct experiments.

## Data availability

All data supporting the conclusions of this study are available within the article and its supplementary figures and data.

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

## Acknowledgements
We thank members of the Machner laboratory for their critical reading of the manuscript, Dr. Joseph T. Wade (New York State Department of Health, Wadsworth Center) for advice and direction on *boxA* motifs, and Dr. Caroline Esnault (National Institutes of Health) for instruction on qPCR procedure. This work was funded by the Intramural Research Program of the National Institutes of Health, USA (Project Number: 1ZIAHD008893-10).

## Author contributions
N.A.E., B.K. and M.P.M. contributed to experimental design; N.A.E., B.K. and J.T. performed experiments; N.A.E. and M.P.M. composed the manuscript.

## Competing interests
The authors declare no competing interests.
