## [Peer Review File · Communications Biology]

Reviewers' Comments:

Reviewer #1:

Remarks to the Author:

copied in by editor from attachment

Summary

This manuscript describes the development of a multiplex CRISPRi platform in the pathogen *Legionella pneumophila*. To this end, the authors adopted a strong processive promoter with a boxA element upstream of a repeat/spacer array. The authors claimed that the boxA element inserted into the front of the repeat prevented transcriptional termination in the middle of the long crRNA. As the authors know very well, the CRISPRi systems have been used for genome regulation in many kinds of microbes. And now they are common tools for researchers to conduct various biological and molecular studies using conventional or unconventional microbes. I did have trouble determining exactly what has been provided for something new of the multiplex CRISPRi because there is little evidence that boxA prevents long crRNA from anti-termination. In addition, there is a lack of examples of the need to conduct 10 or multiplex repression. As an example of multiplex repression, figure 6 shows the control of CRISPRi activity for each spacer position, but the predictability is poor and the cloning steps are very complex. A method introducing mismatches into one gRNA is much simpler, more efficient, and less expensive than the method the authors presented in this study. However, if the authors provide stronger evidences of the boxA effect on the multiplex CRISPRi efficiency, it would be reconsidered for publication.

Major comment

1. Why is the standard deviation large in the same sample in Figure 3B, Figure 3D, Figure 4B and Figure 6B? Are each result statistically significant?
2. In Figure 3B, for lidA, dotD crRNA, why is CRISPRi activity higher in S9 or S10 than in S7 or S8?
3. Since qRT-PCR data of Figure 3B and Figure 4B (boxA-less) are used repeatedly, it must be specified in the legend. If duplicated, the samples of the two figures would be the same and then why the western blot pictures in Figure 3C and Figure 4B (boxA-less) are different? The western blot result in Figure 3C looks different from the that in Figure S5.
4. In Figure 4, direct evidence that boxA prevents transcription termination of crRNA is essential. In the absence of tracrRNA (prevention of crRNA processing), the qRT-PCR data
Journal: Communications Biology
Manuscript ID: COMMSBIO-20-1719-T
Title: A multiplex CRISPR interference tool for virulence gene interrogation in an intracellular pathogen
of upstream and downstream of the precursor crRNA should be added in the absence or presence of boxA.
5. In Figure 4B, why is there no boxA effect for lidA and vipD? The reason mentioned by the author does not appear to be appropriate. Why is CRISPRi activity low for S1 or S2 of dotO boxA(-90)? Can you statistically conclude that there is an overall boxA effect?
6. In Figure 6B, the repression of the S4 spacer is stronger than that of S2. Why does growth in 6D do better than S2? Why does the cell growth in 6D looks tunable, but 6B looks like on/off?
7. It is rare to repress up to 10 multiple genes at the same time, so the author should provide examples that show significant effects by simultaneously repressing more than 5 different genes.
8. There are 11 direct repeats of 36-bp when operating in 10-plex. Are there any problems such as deletion due to direct repeat? Isn't the standard deviation of each sample large

due to this problem? You should check the correctness of the crRNA cassette used in each experiment. Also, in Figure S1, the MC array was DNA synthesized in GenScript, but is it expensive and time consuming because there are many repeats?

Reviewer #2:

Remarks to the Author:

Summary

Ellis et al. develop an array-based CRISPRi system for the intracellular pathogen *Legionella pneumophila* that has considerable multiplexing capabilities. They optimize CRISPRi by using a combination of a strong, inducible promoter (Ptet) and including a boxA sequence in the leader sequence that enables knockdown from distal spacers in arrays of up to ten spacers total. The authors then show that their system allows for modulation of pathogenesis in a human macrophage model of infection, cleverly using spacer position within the CRISPR array to vary knockdown level (with an array lacking a boxA sequence). Overall, the manuscript represent a significant advance in the use of CRISPR technology on multiple levels:

- 1) These are the longest arrays I've seen yet, and they are applied to a biological problem that might actually require this much multiplexing to solve (the ~300 effectors in *L. pneumophila*).
- 2) This work uses the natural tracrRNA and repeat framework effectively, in contrast to most work that uses unstable sets of single guide RNAs (sgRNAs).
- 3) Most people use sgRNAs to get around problems with expressing distal spacers in arrays—the authors nicely address this problem by increasing RNA polymerase (RNAP) processivity via boxA. This work is likely to have an outsized impact on research in CRISPR tools/engineering and pathogenesis, both of which have focused on sgRNAs (rather than arrays) and mostly targeted pairs of genes when multiplexing. Assuming the strategies employed here are generalizable, many investigators may switch to arrayed multiplexing in their organisms of interest.

General Comments

1) Transcriptional phenomena are poorly described throughout the paper. First, guides that target the non-template strand are more effective (as many have seen before), but the authors make a confusing statement about the PAM and the template strand (line 107) that I imagine readers will interpret to mean the opposite. The authors also seem to inadvertently suggest that strong promoters drive RNAP processivity (line 136), but what is more likely is that some fraction of the elongating RNAP is terminated before the end of the array and increasing the total number of initiating RNAPs simply means that more RNAPs will make it to the end of the array even though the fraction terminated is constant.

2) I think the authors should avoid the conclusion that Rho-dependent termination is leading to shorter array transcripts—at least walk back the language of certainty used in the abstract, i.e., “Constraints on precursor-crRNA expression by Rho-dependent transcription termination were overcome”. boxA sequences have multiple effects on RNAP: 1) they increase the speed of transcription, 2) they prevent RNAP from forming arrested (or backtracked) complexes, and 3) they block both intrinsic and Rho-dependent termination (these three enhancements are not necessarily mutually exclusive)—any of these effects could result in more full-length array RNAs. Although I agree that Rho is a likely culprit, the authors have not shown that Rho is important experimentally—nor should they have to if they walk back their claim. I know that the Stringer et al. manuscript cited by the authors has decent evidence that the effect is Rho-mediated, but that manuscript has not been peer reviewed. I would just say that boxA improves RNAP processivity by multiple mechanisms in the abstract/results of the paper and suggest Rho termination as the cause in the discussion. Whether or not the boxA effect is cause by Rho is actually only a very minor point in an otherwise strong paper,

so there is no need to overclaim about it.

3) For Fig. 6, I think it's strange to claim that multiplex CRISPRi works during host infection when only one spacer targets a virulence factor. Maybe make a claim about the knockdown titration as the figure title instead.

Specific Comments

Fig. 4b. I think something is wrong with the dotD panel. qPCR shows substantial repression of dotD, but the immunoblots all look the same. Is it possible that the blot is actually the loading control, rather than dotD?

Reviewer #3:

Remarks to the Author:

The manuscript by Ellis et. al. is a tool development paper that employs multiplex CRISPRi to silence up to 10 genes in the pathogen *Legionella pneumophila*. I am very enthusiastic about this paper because *Legionella* effectors have been proven to be redundant and difficult to study, and this paper has the potential to be a valuable resource to assign functions to individual effectors. These were my concerns/comments:

- 1) A little more detail into the robustness of the system would be useful. For e.g, what are the temporal limitations? How soon after Tetracycline treatment can they see gene disruption? The authors do an overnight induction with 20 ng/mL or 40 ng/mL anhydrous tetracycline. What if they add tetracycline after infection in macrophages? For researchers studying temporal kinetics of multiple effectors, this system of knocking down individual effectors at various times during infection could be very useful.
- 2) What is the readout of their gene disruption when their array consists of effectors that are in the same operon or clustered together vs when they are in different positions of the genome? For e.g., it is a bit concerning that they did not achieve suitable disruption of certain genes regardless of their positions in the cassette.
- 3) The authors wrote, "The presence of boxA in either the -58 or -90 position maximized silencing of sidM, dotO, dotD, ravN, and lpg2793 by crRNAs encoded from all spacer positions..". However, I do not find the data on DotD silencing when boxA was added in -58 or -90 position convincing (after looking at the western blots).

Minor point: Name the y-axis in figure 6B.

Response to the Reviewers

Manuscript: COMMSBIO-20-1719-T

A multiplex CRISPR interference tool for virulence gene interrogation in an intracellular pathogen

Nicole A. Ellis, Byoungkwan Kim, Jessica Tung, and Matthias P. Machner*

*Correspondence:

Phone: (301) 435-8417

[Email: machnerm@nih.gov](mailto:machnerm@nih.gov)

Reviewers' comments:

Reviewer #1 (Remarks to the Author):

copied in by editor from attachment

Summary

This manuscript describes the development of a multiplex CRISPRi platform in the pathogen *Legionella pneumophila*. To this end, the authors adopted a strong processive promoter with a boxA element upstream of a repeat/spacer array. The authors claimed that the boxA element inserted into the front of the repeat prevented transcriptional termination in the middle of the long crRNA. As the authors know very well, the CRISPRi systems have been used for genome regulation in many kinds of microbes. And now they are common tools for researchers to conduct various biological and molecular studies using conventional or unconventional microbes. I did have trouble determining exactly what has been provided for something new of the multiplex CRISPRi because there is little evidence that boxA prevents long crRNA from anti-termination. In addition, there is a lack of examples of the need to conduct 10 or multiplex repression.

*Reviewer 2 nicely outlined the many novelties of our study. We chose to build this technology platform in *Legionella pneumophila* so that it could be applied to understand the, often redundant, role of the >300 different virulence genes, a need appreciated by both Reviewers 2 and 3. Beyond probing the pool of *Legionella pneumophila* virulence genes, other applications could include investigating the regulation of cyclic-di GMP levels in bacteria, for example, as bacterial genomes often encode dozens of proteins with cyclic-di GMP synthesis and hydrolysis capabilities (e.g. *Pseudomonas aeruginosa* encodes 40 (Ryan et. al J. Bacteriol 2006), *Legionella pneumophila* encodes 22 (Levi et. al mBio 2011), and *Vibrio cholerae* encodes at least 11 (Romling et. al Microbiol Mol Biol Rev 2013)). Furthermore, in the discussion, we provided examples for how an array bearing 10 spacers could be used in different ways to maximally silence less than 10 targets as well (lines 366-368).*

As an example of multiplex repression, figure 6 shows the control of CRISPRi activity for each spacer position, but the predictability is poor and the cloning steps are very complex.

We would argue that spacer position, knockdown, and phenotype were very predictable for a gene silencing tool, as further addressed in point #6 below. Numerous data points are shown in each figure to provide the most transparent view of the capabilities of our platform. Our cloning strategy may at first appear complex, though it applies the standardized, commercially available Invitrogen Gateway cloning strategy for the movement of DNA fragments from donor to destination plasmids.

A method introducing mismatches into one gRNA is much simpler, more efficient, and less expensive than the method the authors presented in this study.

We do not claim that our approach should replace mismatch CRISPRi as a gene dosage tool since each approach has its advantages. In fact, Reviewer 2 feels that repeat/spacer arrays have advantages to unstable sgRNAs. Our technology was not tested “head-to-head” with any gRNA platform, but rather provides an alternative option for researchers.

However, if the authors provide stronger evidences of the boxA effect on the multiplex CRISPRi efficiency, it would be reconsidered for publication.

We followed the reviewer’s recommendation and added two new experiments that address those concerns. First, we performed an experiment to directly test the level of precursor-crRNA expression in the presence and absence of boxA, see point #4 below. We found that the levels are increased by at least an order of magnitude in constructs containing boxA elements as opposed to constructs lacking boxA (Figure 4). Moreover, we now show that boxA-containing 10-plex constructs accomplish gene silencing over a period of 48 hours of infection, even with crRNAs encoded at the S9/10 position (Figure 7). Together, these findings provide strong evidence that boxA-containing 10-plex CRISPRi is capable of silencing entire groups of genes.

Major comment

1. Why is the standard deviation large in the same sample in Figure 3B, Figure 3D, Figure 4B and Figure 6B? Are each result statistically significant?

We remind Reviewer 1 that knockdown (vs. knock-out) technologies are intrinsically noisy, especially when relying on induction of a system. One could draw comparisons between CRISPRi knockdown and the well acknowledged noise of RNAi knockdown. There is biological/cellular noise intrinsic to gene expression. Here we performed multiple replicates for each experiment and are showing those individual data points to be up-front about the noise of knockdown. We also remind the reviewer that the knockdown experiments shown here are intended to be initial discovery experiments to be followed up with true deletions (as noted in lines 414-417) in which more reproducible data are expected.

2. In Figure 3B, for lidA, dotD crRNA, why is CRISPRi activity higher in S9 or S10 than in S7 or S8?

We, like Reviewer 1, are puzzled that the dotD repression data for S9 or S10 do not fit the general downward trend that we see for the other targets. Without further investigation of this outlier, we do not have a good explanation for this phenomenon, although it appears to be unique for this target gene. Protein profiles did suggest decreased repression. We have now acknowledged this in the manuscript:

“dotD showed better than expected gene silencing with the most distal S9 spacer (MC-IV) for reasons that remain to be determined.” (lines 178-179)

Regarding lidA, we would argue that the difference in CRISPRi activity in S9 or S10 and S7 or S8 is negligible.

3. Since qRT-PCR data of Figure 3B and Figure 4B (boxA-less) are used repeatedly, it must be specified in the legend. If duplicated, the samples of the two figures would be the same and then why the western blot pictures in Figure 3C and Figure 4B (boxA-less) are different? The western blot result in Figure 3C looks different from the that in Figure S5.

While we had specified the use of Figure 3 mRNA data in the Figure 4 legend, we have re-worded the Figure 4 legend in hopes this brings clarity to readers (updated Figure 4 legend is shown below alongside the new Figure 4 in point #4).

Since immunoblot results are visualized, new samples for side-by-side analysis of the boxA-less and boxA-containing samples were collected, therefore, Figure 3C and what is now Figure 4D should look different. Furthermore, the immunoblots in Figure 3C correspond to Figure S3 and the immunoblots in Figure 4D correspond to Figure S5 as each figure legend indicates.

4. In Figure 4, direct evidence that boxA prevents transcription termination of crRNA is essential. In the absence of tracrRNA (prevention of crRNA processing), the qRT-PCR data of upstream and downstream of the precursor crRNA should be added in the absence or presence of boxA.

We thank Reviewer 1 for the suggestion. We constructed MC-II- and MC-IV-expressing plasmids which lack the tracrRNA-encoding sequence. We induced expression of the precursor-crRNA and compared the levels of this RNA at three different locations along the precursor-crRNA in boxA-less, boxA(-58), and boxA(-90) strains by qPCR. As shown below, and now in the manuscript as Figure 4C, boxA-containing constructs do in fact express an average ~10-fold more precursor-crRNA than boxA-less constructs throughout the entire length of the synthetic repeat/spacer array, suggesting that boxA does promote array transcription.

Figure 4: The effect of *boxA* elements on gene silencing by *P_{tet}*-MC arrays.

(A) Sequence alignment of the *boxA* elements identified in *Coxiella* (16S RNA encoding sequence), *L. pneumophila* sp. *Lens* (CRISPR array) and *E. coli* (*boxA* consensus). (B) The *boxA* motif from the *L. pneumophila* sp. *Lens* CRISPR array was added to the leader sequence of the *P_{tet}*-MC constructs either -58 (*boxA*(-58)) or -90 (*boxA*(-90)) base pairs upstream of the repeat/spacer array. *boxA* is depicted as a pink square. (C) Constructs lacking the *tracr*RNA-encoding sequence were used to probe precursor-crRNA expression from *boxA*-containing vs. *boxA*-less constructs in axenic cultures of *Lp02(dcas9)* (+40 ng/mL aTC). Precursor-crRNA levels were measured by qPCR using primer pairs that annealed to three different regions of the precursor-crRNA. Fold increased expression vs. *boxA*-less constructs was determined using $\Delta\Delta C_T$. Data are ordered by spacer position in constructs MC-II or MC-IV. (D) Constructs with the *tracr*RNA-encoding sequence were used to probe gene repression by qPCR of RNA from axenic cultures of *Lp02(dcas9)* bearing *boxA* constructs to that with the empty vector (+40 ng/mL aTC). Fold repression was determined using $\Delta\Delta C_T$. For comparison, *boxA*-less qPCR data from Figure 3 is shown again. For visualization of protein levels, pellets from *boxA*-less and *boxA*(-58) and *boxA*(-90) strains were collected

and processed for immunoblot analysis. The bands were reordered corresponding to their spacer position. Original immunoblots and ICDH loading controls are shown in Figure S5.

5. In Figure 4B, why is there no boxA effect for lidA and vipD? The reason mentioned by the author does not appear to be appropriate. Why is CRISPRi activity low for S1 or S2 of dotO boxA(-90)? Can you statistically conclude that there is an overall boxA effect?

We cannot provide an explanation for every outlier. We do believe we provide enough examples of boxA having a beneficial effect on gene repression through crRNAs expressed from distal spacers that boxA can be considered generally advantageous. There is no one-fits-all solution for every target as each target has its own intrinsic regulation and genomic environment. We apologize to Reviewer 1 that we cannot be more specific of the details of the mechanism preventing a boxA effect on lidA and vipD.

6. In Figure 6B, the repression of the S4 spacer is stronger than that of S2. Why does growth in 6D do better than S2? Why does the cell growth in 6D looks tunable, but 6B looks like on/off?

We agree with Reviewer 1 that fold repression between S2 and S4 is very similar. Yet, we would argue that the remaining spacers S6, S8, and S10 appear tunable, not on/off (and correlate to a tunable phenotype). The average phenotype data in 6D for S2 and S4 suggests that there is a difference, but upon close examination there are S4 data points that overlap with S2 data points. We again would argue biological noise in an experiment as complex as intracellular growth and infection leads to minor discrepancies.

7. It is rare to repress up to 10 multiple genes at the same time, so the author should provide examples that show significant effects by simultaneously repressing more than 5 different genes.

We would argue that the rarity of examples where 10 genes are simultaneously repressed is because of the lack of means to do so. Deleting 10 genes simultaneously would be a very timely and laborious process, especially in a genetically less tractable organism. We hope this technology will help to elucidate additional examples.

8. There are 11 direct repeats of 36-bp when operating in 10-plex. Are there any problems such as deletion due to direct repeat? Isn't the standard deviation of each sample large due to this problem? You should check the correctness of the crRNA cassette used in each experiment. Also, in Figure S1, the MC array was DNA synthesized in GenScript, but is it expensive and time consuming because there are many repeats?

We would like to reassure Reviewer 1 that all constructs used in this study were sequenced for correctness at each step of the cloning process. We did not observe any deletion of spacers from the array. While synthesizing arrays from GenScript is expensive, we do not think expenses invalidate our technology. In fact, our advanced protocol for the assembly of CRISPR arrays from synthetic oligonucleotide pairs will all but eliminate the need to purchase arrays in the future.

Reviewer #2 (Remarks to the Author):

Summary

Ellis et al. develop an array-based CRISPRi system for the intracellular pathogen *Legionella pneumophila* that has considerable multiplexing capabilities. They optimize CRISPRi by using a combination of a strong, inducible promoter (Ptet) and including a boxA sequence in the leader sequence that enables knockdown from distal spacers in arrays of up to ten spacers total. The authors then show that their system allows for modulation of pathogenesis in a human macrophage model of infection, cleverly using spacer position within the CRISPR array to vary knockdown level (with an array lacking a boxA sequence). Overall, the manuscript represents a significant advance in the use of CRISPR technology on multiple levels:

1) These are the longest arrays I've seen yet, and they are applied to a biological problem that might actually require this much multiplexing to solve (the ~300 effectors in *L. pneumophila*).

2) This work uses the natural tracrRNA and repeat framework effectively, in contrast to most work that uses unstable sets of single guide RNAs (sgRNAs).

3) Most people use sgRNAs to get around problems with expressing distal spacers in arrays—the authors nicely address this problem by increasing RNA polymerase (RNAP) processivity via boxA.

This work is likely to have an outsized impact on research in CRISPR tools/engineering and pathogenesis, both of which have focused on sgRNAs (rather than arrays) and mostly targeted pairs of genes when multiplexing. Assuming the strategies employed here are generalizable, many investigators may switch to arrayed multiplexing in their organisms of interest.

We thank Reviewer 2 for recognizing the contributions this manuscript makes to the field of CRISPR tools/engineering.

General Comments

1) Transcriptional phenomena are poorly described throughout the paper. First, guides that target the non-template strand are more effective (as many have seen before), but the authors make a confusing statement about the PAM and the template strand (line 107) that I imagine readers will interpret to mean the opposite.

We appreciate Reviewer 2 pointing out this confusing statement and have revised the text to read:

“In agreement with other studies (12, 14, 28), we found that targeting a 30 base pair sequence near the start of the gene on the non-template strand provided the most efficient knockdown (Figure 1C, arrowheads C and D). All target sequences were upstream of a PAM sequence (“NGG” on the template strand (29)).” (lines 106-109)

The authors also seem to inadvertently suggest that strong promoters drive RNAP processivity (line 136), but what is more likely is that some fraction of the elongating RNAP is terminated before the end of the array and increasing the total number of initiating RNAPs simply means that more RNAPs will make it to the end of the array even though the fraction terminated is constant.

We thank Reviewer 2 for clarifying the mechanisms of RNAP so that we may make this the most accurate manuscript. We have revised the text to read:

*“using the native *S. pyogenes* promoter (P_{native}) to express the multiplex CRISPR (MC) repeat/spacer arrays proved insufficient in repressing more than three to four targets, possibly due to fewer incidences of transcription initiation at a weaker promoter and therefore, fewer opportunities for RNA polymerase to complete transcription of the entire array before termination (Figure 2B).” (lines 134-137)*

2) I think the authors should avoid the conclusion that Rho-dependent termination is leading to shorter array transcripts—at least walk back the language of certainty used in the abstract, i.e., “Constraints on precursor-crRNA expression by Rho-dependent transcription termination were overcome”. boxA sequences have multiple effects on RNAP: 1) they increase the speed of transcription, 2) they prevent RNAP from forming arrested (or backtracked) complexes, and 3) they block both intrinsic and Rho-dependent termination (these three enhancements are not necessarily mutually exclusive)—any of these effects could result in more full-length array RNAs. Although I agree that Rho is a likely culprit, the authors have not shown that Rho is important experimentally—nor should they have to if they walk back their claim. I know that the Stringer et al. manuscript cited by the authors has decent evidence that the effect is Rho-mediated, but that manuscript has not been peer reviewed.

I would just say that boxA improves RNAP processivity by multiple mechanisms in the abstract/results of the paper and suggest Rho termination as the cause in the discussion. Whether or not the boxA effect is caused by Rho is actually only a very minor point in an otherwise strong paper, so there is no need to overclaim about it.

We thank Reviewer 2 for the suggestion on how to better describe the contribution of boxA to the expression of our CRISPR arrays. We have made changes throughout the text accordingly.

Examples of new text include:

Abstract: “Constraints on precursor-crRNA expression were overcome by combining a strong promoter with a boxA element upstream of a repeat/spacer array.” (lines 20-22)

Results: “BoxA elements are often found upstream of long non-coding RNAs, like the 16S RNA (35), where they are bound by the Nus complex to improve RNA polymerase processivity by multiple mechanisms (36).” (lines 202-204)

Discussion: “Several bacterial CRISPR systems appear to have overcome polarity of spacer expression through acquiring boxA sequences upstream of their arrays (37). boxA could improve RNA polymerase processivity of these and our synthetic arrays (Figure 4) by a number of different mechanisms, one of which may be blocking Rho-dependent transcription termination (52-54), as hypothesized by Stringer et al (37).” (lines 389-393)

3) For Fig. 6, I think it's strange to claim that multiplex CRISPRi works during host infection when only one spacer targets a virulence factor. Maybe make a claim about the knockdown titration as the figure title instead.

We thank Reviewer 2 for the suggestion to change the title of Figure 6 which now reads:

“Gene repression during infection is titratable with multiplex CRISPRi.” (line 796)

We have also now added Figure 7 to the paper in which we do in fact show knockdown of at least eight (randomly selected) genes throughout the duration of a 48-hour infection using a MC 10-plex construct. This should provide the final missing piece of evidence that multiplex CRISPRi can be used to investigate gene function in this context. See below (Figure 7 in the manuscript).

Figure 7: Multi-gene knockdown by *boxA*-MC persists throughout infection.

(A) *boxA(-58)* MC 10-plex was constructed with ten unique spacers for expression of crRNAs targeting ten unique predicted effector-encoding genes. (B) *Lp02(dcas9)* bearing either the empty vector or *boxA(-58)* MC 10-plex was used to infect U937 macrophages, and intracellular growth was determined by measuring the average of CFU/well of two to three experiments at 2 and 48 hours post infection (hpi). (C) At 48 hpi bacteria were extracted from the U937 macrophages and harvested for RNA. mRNA levels of each of the ten target genes was measured by qPCR. Fold repression of target genes in *boxA(-58)* MC 10-plex compared to vector-bearing *Lp02(dcas9)* was determined using $\Delta\Delta C_T$.

Specific Comments

Fig. 4b. I think something is wrong with the dotD panel. qPCR shows substantial repression of dotD, but the immunoblots all look the same. Is it possible that the blot is actually the loading control, rather than dotD?

To address Reviewer 2's (and Reviewer 3's below) concerns about the dotD immunoblot, we re-examined our samples by loading less material onto the gel to better resolve individual bands. Samples were also run in order of the spacer position (vs. construct order) to eliminate the need of having to crop the bands. We believe the new immunoblots which have been incorporated into Figure 4 (shown below) and Supplemental Figure 5 show the decrease in repression at later spacer position for *boxA*-less constructs and increased repression by all spacer positions for *boxA(-58)* construct, which is in agreement with the qPCR data. Admittedly *boxA(-90)* data shows more noise in knockdown. While the qPCR data shows extreme levels of gene repression, we believe that slower protein turnover rates/increased protein stability of dotD compared to other proteins probed must drive the slight discrepancy between the qPCR and immunoblot data. We have added a brief statement about this to the main text:

“Minor discrepancies in dotD and DotD levels suggest slower protein turnover rates/increased protein stability of DotD compared to other proteins probed.” (lines 236-238)

**Reviewer #3 (Remarks to the Author):**

The manuscript by Ellis et. al. is a tool development paper that employs multiplex CRISPRi to silence up to 10 genes in the pathogen Legionella pneumophila. I am very enthusiastic about this paper because Legionella effectors have been proven to be redundant and difficult to study, and this paper has the potential to be a valuable resource to assign functions to individual effectors. These were my concerns/comments:

*We thank Reviewer 3 for recognizing the contributions this technology can have to the study of Legionella pneumophila pathogenesis.*

1) A little more detail into the robustness of the system would be useful. For e.g, what are the temporal limitations? How soon after Tetracycline treatment can they see gene disruption? Th authors do an overnight induction with 20 ng/mL or 40 ng/mL anhydrous tetracycline.

*We thank the reviewer for the suggestion of a time-course experiment (see results below). As our induction protocol stands, we induce and subculture the Legionella over a period of two days to obtain what we consider the "silenced, final strains" used for infection. Unfortunately, when Legionella undergoes its bi-phasic lifestyle during axenic growth, the effector-encoding genes are naturally up- and down-regulated dependent on the growth stage. This oscillation in gene expression caused the level of SidM and RavN to be downregulated in exponential phase (samples 2-4) even in the vector control strain. Upon reaching late exponential phase (sample 5), the proteins re-appeared in the control strain but remained silent in the CRISPRi strain.*

*Thus, the natural change in gene expression prevented us from accurately determining the silencing efficiency by CRISPRi early in induction, and the only informative time points to consider are samples 1 vs. 5 where sample 1 is before induction of the CRISPR array and sample 5 is after the two day induction has occurred and effector-encoding gene expression is occurring.*

*Since this experiment did not provide any meaningful data, we decided to not to show it in the manuscript.*

What if they add tetracyclin after infection in macrophages? For researchers studying temporal kinetics of multiple effectors, this system of knocking down individual effectors at various times during infection could be very useful.

There are ample applications of CRISPRi with respect to the timing of induction/gene knockdown as it pertains to context of the host infection. We would like to emphasize though that CRISPRi-mediated gene silencing, though efficient, does not result in an instantaneous loss of the encoded protein. Proteins will disappear only gradually either through proteolytic degradation or dilution upon bacterial cell division. We seek to explore those possibilities in the near future, but at this time, it is out of the scope of this paper which aims to establish feasibility of the CRISPRi platform.

2) What is the readout of their gene disruption when their array consists of effectors that are in the same operon or clustered together vs when they are in different positions of the genome?

We agree with the reviewer that silencing the first gene in an operon can have polar effects on downstream genes. In some cases, such as the bulk silencing of redundant effector-encoding genes, we actually view disrupting operons as advantageous – that is knocking down expression of as many genes as possible to observe a phenotype. Each operon will prove to be a special case in which only follow-up studies with in-frame gene deletions will be able to resolve the specific contributions of each gene to the phenotype. We like to think of multiplex CRISPRi as a discovery tool rather than a fine-tuned scalpel to dissect gene function. We have added a comment to the main text mentioning the possibility of polar effects within operons to the discussion (lines 414-417). In terms of Legionella pneumophila, we believe that there are only a few effector-encoding genes actually in operons.

For e.g., it is a bit concerning that they did not achieve suitable disruption of certain genes regardless of their positions in the cassette.

We remind Reviewer 3 that any gene silencing technology that relies on interference is very target specific. Each target has its own intrinsic regulation and genomic context. Should a researcher have a specific set of genes of interest, crRNAs can be screened for most efficient silencing. We believe that it is good to show readers that, similar to RNA interference in mammalian cells, not every crRNA is going to achieve the desired effect.

3) The authors wrote, “The presence of boxA in either the -58 or -90 position maximized silencing of sidM, dotO, dotD, ravN, and lpg2793 by crRNAs encoded from all spacer positions..”. However, I do not find the data on DotD silencing when boxA was added in -58 or -90 position convincing (after looking at the western blots).

Refer to specific comments by Reviewer 2 above. New immunoblots have been provided for DotD.

Minor point: Name the y-axis in figure 6B.

We have corrected this error in the revised manuscript.

Reviewers' Comments:

Reviewer #1:

Remarks to the Author:

All concerns that I have raised are adequately addressed. I recommend the publication of the revised manuscript.

Reviewer #2:

Remarks to the Author:

The authors have completely addressed my concerns. I appreciate the inclusion of new data-- particularly Fig. 7 showing multiplexed knockdown of genes during infection.